# Impaired myofibroblast proliferation is a central feature of pathologic post-natal alveolar simplification

Imran S Khan[1,2], Christopher Molina[2,3,4], Xin Ren[2,3,4], Vincent C Auyeung[2,3], Max Cohen[3,4], Tatsuya Tsukui[2,3,4], Amha Atakilit[2,3,4], Dean Sheppard[2,3,4]*

[1]Division of Neonatology, Department of Pediatrics, UCSF, San Francisco, United States; [2]Cardiovascular Research Institute, UCSF, San Francisco, United States; [3]Division of Pulmonary, Critical Care, Allergy, and Sleep, UCSF, San Francisco, United States; [4]Department of Medicine, UCSF, San Francisco, United States

*For correspondence:
dean.sheppard@ucsf.edu

## eLife Assessment

This study provides **important** insights into postnatal lung development and the mechanisms underlying bronchopulmonary dysplasia (BPD), a condition with high morbidity and mortality in newborns. Through the use of neonatal hyperoxia, cell-type-specific inactivation of Tgfbr2, and other injury models, the research focuses on the role of TGF-β signaling in BPD pathogenesis, highlighting impaired myofibroblast proliferation as a key factor. The inactivation of Etc2 in Pdgfra-lineaged cells disrupts myofibroblast cytokinesis, leading to alveolar simplification and reduced cell numbers. The use of transgenic mice and single-cell transcriptomics offers a detailed and high-quality dataset, advancing our understanding of BPD and serving as a invaluable resource for developmental biology and neonatal pulmonary research. The study's comprehensive approach, robust data, and methodological rigor make it a **compelling** contribution to the field, providing both mechanistic insights and a resource for further research into BPD pathogenesis.

**Abstract** Premature infants with bronchopulmonary dysplasia (BPD) have impaired alveolar gas exchange due to alveolar simplification and dysmorphic pulmonary vasculature. Advances in clinical care have improved survival for infants with BPD, but the overall incidence of BPD remains unchanged because we lack specific therapies to prevent this disease. Recent work has suggested a role for increased transforming growth factor-beta (TGFβ) signaling and myofibroblast populations in BPD pathogenesis, but the functional significance of each remains unclear. Here, we utilize multiple murine models of alveolar simplification and comparative single-cell RNA sequencing to identify shared mechanisms that could contribute to BPD pathogenesis. Single-cell RNA sequencing reveals a profound loss of myofibroblasts in two models of BPD and identifies gene expression signatures of increased TGFβ signaling, cell cycle arrest, and impaired proliferation in myofibroblasts. Using pharmacologic and genetic approaches, we find no evidence that increased TGFβ signaling in the lung mesenchyme contributes to alveolar simplification. In contrast, this is likely a failed compensatory response, since none of our approaches to inhibit TGFβ signaling protect mice from alveolar simplification due to hyperoxia while several make simplification worse. In contrast, we find that impaired myofibroblast proliferation is a central feature in several murine models of BPD, and we show that inhibiting myofibroblast proliferation is sufficient to cause pathologic alveolar simplification. Our results underscore the importance of impaired myofibroblast proliferation as a central feature of alveolar simplification and suggest that efforts to reverse this process could have therapeutic value in BPD.

## Introduction

During the alveolar phase of lung development, alveolar walls protrude from secondary crests into the lumen of existing airways and form new alveoli to increase the lung's surface area for gas exchange (*Jobe, 1999*). Many premature infants are born prior to the onset of alveologenesis and develop bronchopulmonary dysplasia (BPD), a chronic lung disease of prematurity caused by developmental arrest of secondary alveolar septation (*Thébaud et al., 2019*). Over 50 years ago, Northway and colleagues described BPD as a heterogeneous pattern of severe lung injury with pathologic findings of severe inflammation, airway dysplasia, and fibrosis (*Northway et al., 1967*). Despite advances in neonatal care, the incidence of BPD remains unchanged as 30% of infants born before 30 weeks of gestation will develop BPD (*Thébaud et al., 2019*; *Oak and Hilgendorff, 2017*). More infants are now surviving at earlier gestational ages, and the lungs of these infants have a 'new' BPD phenotype characterized by a homogenous process of alveolar simplification, dysmorphic pulmonary vasculature, and mild airway thickening and fibrosis (*Jobe, 1999*; *Thébaud et al., 2019*). BPD is defined clinically by the need for prolonged respiratory support and supplemental oxygen, and infants with BPD suffer from long-term morbidity including severe respiratory infections, reactive airway disease, pulmonary hypertension, and neurodevelopmental impairment (*Jobe, 1999*; *Thébaud et al., 2019*).

The inflammatory response plays a significant role in BPD pathogenesis and results from a combination of perinatal infection, oxygen toxicity, and barotrauma from mechanical ventilation (*Jobe, 1999*). Several recent studies have suggested a role for the pro-fibrotic cytokine transforming growth factor-beta (TGFβ) in BPD pathogenesis (*Oak and Hilgendorff, 2017*; *Saito et al., 2018*; *Surate Solaligue et al., 2017*; *Mižíková and Morty, 2015*; *Alejandre-Alcázar et al., 2007*; *Nakanishi et al., 2007*). Infants with BPD have increased TGFβ in serum and bronchoalveolar lavage fluid (*Vento et al., 2006*; *Jónsson et al., 2000*; *Kotecha et al., 1996*; *Lecart et al., 2000*). Experimentally, murine models of BPD show increased TGFβ signaling and activation of downstream fibrotic pathways (*Mižíková and Morty, 2015*; *Alejandre-Alcázar et al., 2007*; *Nakanishi et al., 2007*; *Mižíková et al., 2015*; *Witsch et al., 2014a*; *Witsch et al., 2014b*; *Rath et al., 2017*). At the same time, TGFβ plays an important role in normal development, and complete loss of TGFβ signaling impairs either embryonic lung development or alveolar septation (*Ahlfeld et al., 2016*; *Chen et al., 2005*; *Kulkarni et al., 1993*; *Li et al., 2016*; *Li et al., 2008*; *Miao et al., 2021*; *Noe et al., 2019*; *Plosa et al., 2014*; *Sureshbabu et al., 2015*; *Colarossi et al., 2005*; *Chen et al., 2008*; *Gao et al., 2022*). TGFβ is a known regulator of myofibroblasts in fibrotic disease (*Frangogiannis, 2020*), but it also drives mesenchymal cells to commit towards the myofibroblast lineage during embryonic lung development (*Li et al., 2016*). Despite its essential role in normal lung development, the functional significance of increased TGFβ signaling in BPD pathogenesis remains unclear.

Alveolar myofibroblasts are contractile mesenchymal cells that are critical for alveolar septation (*Li et al., 2018*). They are characterized by the expression of platelet-derived growth factor receptor alpha (PDGFRα), alpha-smooth muscle actin (α-SMA), and the production of elastin (*Li et al., 2016*; *Li et al., 2018*; *Li et al., 2015*; *Ushakumary et al., 2021*; *Branchfield et al., 2016*). Myofibroblasts are derived from PDGFRα+ mesenchymal cells at birth and peak during alveologenesis when they colocalize with secondary septae (*Li et al., 2016*; *Li et al., 2018*; *Ushakumary et al., 2021*; *Branchfield et al., 2016*; *Hagan et al., 2020*). Ablation of PDGFRα+ cells in neonatal mice causes alveolar simplification, while neonatal hyperoxia treatment causes loss of PDGFRα+ fibroblasts, dysregulated alveolar elastin deposition, and impaired PDGFRα+ cell contractility (*Li et al., 2018*; *Ushakumary et al., 2021*; *Branchfield et al., 2016*; *Riccetti et al., 2022*). Inhibition of myofibroblast contraction has been shown to lead to alveolar simplification (*Li et al., 2020*). These studies support a fundamental functional role for myofibroblasts during alveologenesis, but whether the number of myofibroblasts is reduced in BPD or experimental models of BPD remains controversial (*Li et al., 2018*; *Branchfield et al., 2016*; *Benjamin et al., 2007*; *Bozyk et al., 2012*; *Popova et al., 2014*).

To identify conserved mechanism of injury in BPD pathogenesis, we implemented a novel strategy to compare two murine models of alveolar simplification by single-cell RNA-sequencing (scRNA-seq): neonatal hyperoxia exposure and loss of epithelial TGFβ signaling. Using flow cytometry and sequencing, we observed a dramatic reduction of myofibroblasts in both models of disease. Additionally, we found that increased TGFβ signaling, decreased PDGFRα signaling, and impaired proliferation are hallmark features of injured myofibroblasts during alveologenesis. Blocking TGFβ through several different approaches consistently worsened hyperoxia-induced disease, suggesting TGFβ plays an

essential homeostatic function in both normal alveolar development and hyperoxia-induced disease, but that increased TGFβ signaling does not seem to be an important driver of alveolar simplification. We also demonstrated that PDGFRα+ cells undergo robust proliferation during normal alveologenesis, but this proliferation is impaired in both hyperoxia-treated and TGFβ-manipulated mice. We showed that modulation of PDGFRα+ cell proliferation is sufficient to cause alveolar simplification, even in the absence of hyperoxic injury. These studies demonstrate that PDGFRα+ cells are profoundly sensitive to injury during alveologenesis, they require TGFβ signaling, and their proliferation is essential for normal alveolar development.

## Results

### Neonatal hyperoxia exposure and loss of epithelial TGFβ signaling both cause alveolar simplification

To identify conserved features of lung injury in bronchopulmonary dysplasia (BPD), we compared two different mouse models of lung injury known to cause alveolar simplification in mice. We first implemented the hyperoxia-induced model of BPD in which newborn mice are exposed to 75% oxygen from postnatal day (P) 0–10 and then recovered in room air until analysis (*Figure 1A*). This model takes advantage of the fact that mice are born during the saccular phase of lung development and undergo alveolarization from postnatal day P4 – P39 (*Mund et al., 2008*; *Schittny, 2017*; *Vila Ellis and Chen, 2021*). We harvested lungs to analyze alveolar architecture after completion of alveolarization at P40. Hyperoxia-treated mice showed significant alveolar airspace enlargement compared to controls (*Figure 1B*). Consistent with findings from other groups using similar protocols (*Li et al., 2018*; *Branchfield et al., 2016*), we did not see evidence of fibrosis or scarring. In parallel, we used the flexiVent rodent ventilator to measure respiratory mechanics. Hyperoxia-treated mice had increased total lung capacity, increased compliance, and decreased elastance compared to control mice (*Figure 1C*). These findings are consistent with those of emphysematous lungs, and these physiologic changes align with the enlarged, simplified alveolar structures observed by histology (*Liu et al., 2014*; *Tsujino et al., 2017*; *Yee et al., 2009*).

For our second model of alveolar simplification, we generated mice which lack the critical TGFβ receptor TGFBR2 in lung epithelium by crossing Nkx2.1-cre to the *Tgfbr2* conditional allele (*Tgfbr2^{F/F}*;Nkx2.1-cre). These mice are viable at birth, have no respiratory distress, gain normal weight, and generate expected ratios of offspring with respect to each genotype (data not shown). Lungs from conditional knockout (cKO) mice show enlarged airspaces consistent with alveolar simplification, confirming earlier reports that TGFβ signaling in lung epithelium is required for normal secondary septation (*Figure 1D and E*; *Sureshbabu et al., 2015*). We then used the flexiVent system to analyze respiratory mechanics in cKO mice and observed increased compliance and decreased elastance comparable to what we saw in hyperoxia-exposed mice (*Figure 1F*). Previous work demonstrated that *Tgfbr2^{F/F}*;Nkx2.1-cre mice are protected from lung disease caused by transgenic overexpression of TGFβ1 or neonatal hyperoxia treatment (*Sureshbabu et al., 2015*). To our surprise, *Tgfbr2^{F/F}*;Nkx2.1-cre mice were not protected from disease in our model of neonatal hyperoxia, but rather developed worse disease. These data suggest TGFβ signaling to the lung epithelium is required for both normal alveologenesis and that it plays a protective role in the response to hyperoxia-induced injury.

### Profound loss of PDGFRα+ fibroblasts during hyperoxia treatment

To begin exploring the mechanisms underlying alveolar simplification in these models, we treated wildtype mice with hyperoxia from P0-P10 and analyzed dissociated lung cells by flow cytometry at P10. While we observed changes in many cell types of the lung with hyperoxia treatment, the most striking result was the profound loss of PDGFRα+ fibroblasts (*Figure 2A and B*). Using flow cytometry, we found that both the percentage and absolute number of PDGFRα+ fibroblasts were substantially reduced with hyperoxia treatment (*Figure 2B*). Next, we analyzed *Tgfbr2^{F/F}*;Nkx2.1-cre mice at P10 in normoxia and observed a similar phenotype of reduced PDGFRα+ fibroblasts (*Figure 2C and D*). While the magnitude of reduction was less impressive than we observed in the hyperoxia model, the most significant change in cell numbers in *Tgfbr2^{F/F}*;Nkx2.1-cre mice was the reduction in PDGFRα+ fibroblasts. Together, these results are consistent with earlier studies which showed a loss of PDGFRα+

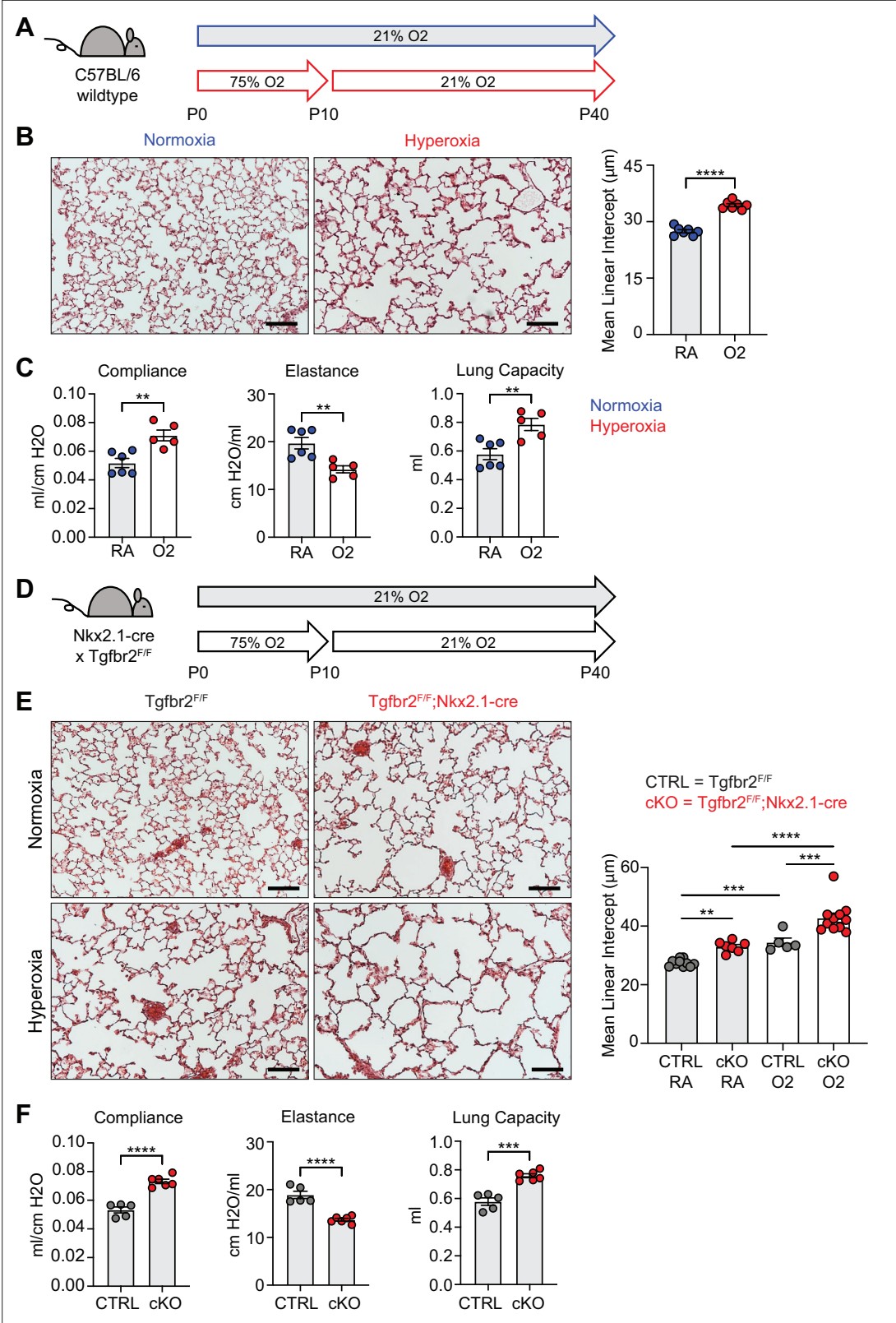

**Figure 1.** Neonatal hyperoxia treatment and loss of epithelial TGFβ signaling both cause alveolar simplification. (**A**) Wildtype C57BL/6 mice were treated in 75% hyperoxia versus normoxia from P0-P10 and recovered in room air until harvest at P40 for analysis by either histology or lung physiology. (**B**) H&E sections of representative lungs from (**A**) harvested at P40 (left) with quantification of mean linear intercept (right). (**C**) Mice treated as in (**A**) and harvested for lung physiology measurements of compliance, elastance and lung capacity. (**D**) *Tgfbr2^{F/F}* and *Tgfbr2^{F/F}*;Nkx2.1-cre littermates were treated

*Figure 1 continued on next page*

*Figure 1 continued*

in hyperoxia versus normoxia from P0-P10 and recovered in room air until harvest at P40 for analysis by histology. (**E**) H&E sections of representative lungs from (**D**) harvested at P40 (left) with quantification of mean linear intercept (right). (**F**) Normoxia cohort treated as in (**D**) and harvested for lung physiology measurements of compliance, elastance and lung capacity. Data in (**B**), (**C**), and (**F**) compared by two-tailed unpaired Student's t-test. Data in (**E**) compared by ANOVA with Fisher's post hoc test. Error bars depict mean ± SEM. \*\*p<0.01, \*\*\*p<0.001, \*\*\*\*p<0.0001. Scale bars = 100 µm.

cells with neonatal hyperoxia treatment and reinforce the critical role of PDGFRα+ fibroblasts in normal alveolar development (*Li et al., 2018*; *Riccetti et al., 2022*).

## scRNA-seq reveals loss of myofibroblasts in both models of alveolar simplification

To identify conserved molecular and cellular mechanisms underlying pathologic alveolar simplification, we performed single-cell RNA sequencing on lungs from *Tgfbr2$^{F/F}$* (CTRL) and *Tgfbr2$^{F/F}$*;Nkx2.1-cre (cKO) mice in either normoxia or hyperoxia (*Figure 3A*). This approach allowed for simultaneous comparison of both injury models: normoxia versus hyperoxia in CTRL mice, and CTRL versus cKO mice in normoxia (*Figure 3B*). Given the reduction of PDGFRα+ fibroblasts observed by flow cytometry in both models of lung injury, we hypothesized that changes in the lung mesenchyme might play a substantial role in the pathogenesis of alveolar simplification. To ensure sufficient mesenchymal cells in our analysis, we used flow cytometric sorting to enrich for mesenchymal and epithelial cells by limiting the input of hematopoietic and endothelial cells. After processing and combining all samples, we identified 24 clusters from 26,610 cells (*Figure 3—figure supplement 1A-C*). We focused our analysis on the lung mesenchyme in which we used the Seurat software package, differential gene expression, and comparison to published scRNA-seq data to assign identities to seven distinct mesenchymal clusters (*Figure 3C* and *Figure 3—figure supplement 2*; *Hurskainen et al., 2021*; *Negretti et al., 2021*; *Narvaez Del Pilar et al., 2022*; *Xia et al., 2023*).

Although flow cytometry had already shown a profound loss of PDGFRα+ cells in these two injury models, we could not attribute this loss to a specific fibroblast subset. Our single-cell data enabled higher resolution assessment of the dynamics of fibroblast subsets (*Figure 3D*). In particular, we observed a significant loss of alveolar and ductal myofibroblast clusters in both models of injury (*Figure 3D and E*), suggesting that the reduction of PDGFRα+ cells by flow cytometry was due to the loss of alveolar and ductal myofibroblasts. To externally validate these findings, we re-analyzed two recently published scRNA-seq studies in which mice were treated with 85% oxygen from P0-P14 (*Hurskainen et al., 2021*; *Xia et al., 2023*). Consistent with our own data, hyperoxia-treated mice in both studies showed a significant loss of myofibroblasts within the lung mesenchyme (*Figure 3—figure supplement 3*).

To gain insight into the molecular mechanisms driving the loss of myofibroblasts, we used Qiagen Ingenuity Pathway Analysis (IPA) to compare differential gene expression in the myofibroblasts in both models of injury (*Krämer et al., 2014*). After filtering for upregulated pathways, we identified 32 predicted upstream regulators that were shared in both models (*Figure 3F*). Of these pathways, we noted several predictions corresponding to TGFβ signaling, inhibitors of cell cycle, and Wnt signaling. These results are notable because TGFβ is a known driver of myofibroblast differentiation and profibrotic programs in fibroblasts (*Li et al., 2016*; *Frangogiannis, 2020*), while its overall contribution to alveolar simplification and BPD is unclear (*Saito et al., 2018*; *Calthorpe et al., 2023*). The prediction of increased cell cycle inhibition is also of interest because it suggests that the reduction of myofibroblasts in these models might be caused by their impaired proliferation.

We used the NicheNet software package to gain insight into the specific cell-cell interactions which might be impacting the myofibroblasts in these two models of lung injury (*Browaeys et al., 2020*). We reasoned that epithelial-myofibroblast signals would be particularly important, since the defect in the cKO model is restricted to epithelial cells, yet the most prominent phenotype is found in myofibroblasts. To explore this further, we used NicheNet to identify shared patterns of changes in ligand-receptor signaling originating from the lung epithelium to the myofibroblasts (*Figure 4A*). Here we again noted an increase in TGFβ signaling to the myofibroblasts (*Figure 4B*). Of the decreased pathways, the most notable were Pdgfa-Pdgfra and Shh-Hhip, both of which are known to be critical for myofibroblast differentiation and function (*Figure 4C*; *Gao et al., 2022*; *Li et al., 2019*). Additionally, the predicted reduction of Pdgfra signaling aligns with our flow cytometry data showing a loss of

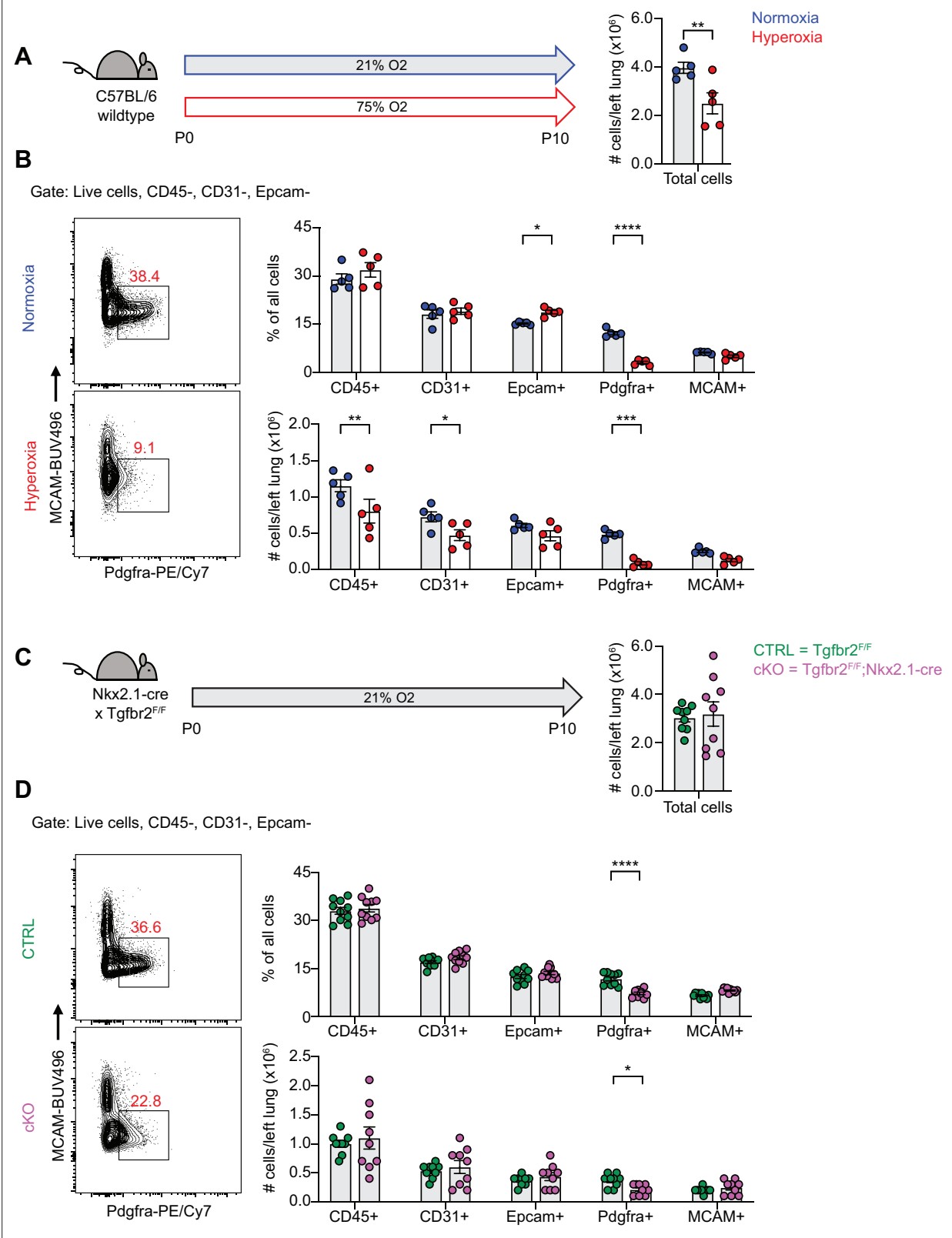

**Figure 2.** Loss of PDGFRα+ Cells with neonatal hyperoxia treatment and loss of epithelial TGFβ signaling. (**A**) Wildtype C57BL/6 mice were treated in 75% hyperoxia versus normoxia from P0-P10 and harvested on P10 for analysis by flow cytometry. Graph on right shows total cells per left lung as quantified by flow cytometry. (**B**) Representative flow cytometry plots of the lung mesenchyme (live, CD45-, CD31-, and Epcam-) with gates depicting PDGFRα+ cells (left). Major cell populations of the lung were defined by the indicated cell surface markers and shown as either a percentage of all

*Figure 2 continued on next page*

*Figure 2 continued*

cells (top) or as absolute number (bottom). (**C**) *Tgfb2<sup>F/F</sup>* and *Tgfbr2<sup>F/F</sup>*;Nkx2.1-cre littermates were maintained in normoxic conditions from P0-P10 and harvested on P10 for analysis by flow cytometry. Graph on right shows total cells per left lung as quantified by flow cytometry. (**D**) Flow cytometry plots for (**C**) using the same gating strategy as described above. Data in (**A–D**) analyzed by two-tailed unpaired Student's t-test. Error bars depict mean ± SEM. *p<0.05, **p<0.01, ***p<0.001, ****p<0.0001.

PDGFRα+ fibroblasts in both injury models as well as a reduction in the mean fluorescence intensity of PDGFRα antibody staining in these cells (*Figure 2B–D*). Together, these data suggest that epithelial dysfunction may cause myofibroblast defects through loss of supportive signals and gain of inhibitory signals.

## TGFβ signaling plays homeostatic role in normal alveolar development and in hyperoxia

Both IPA and NicheNet analyses predicted that TGFβ signaling is activated in myofibroblasts in both models of alveolar simplification (*Figures 3 and 4*). Given the conflicting literature regarding the role of TGFβ in normal lung development (*Saito et al., 2018*; *Ahlfeld et al., 2016*; *Calthorpe et al., 2023*), we sought to determine the functional significance of TGFβ signaling during postnatal alveolo-genesis in both normoxic and hyperoxic conditions. We hypothesized that if excessive TGFβ signaling is a pathologic response in hyperoxia, titration of a pan-TGFβ-blocking antibody (1D11) could identify a therapeutic window to protect from disease while permitting normal development under normoxic conditions. Alternatively, worsened disease with 1D11 treatment would suggest that increased TGFβ is a compensatory response to injury rather than a primary driver of disease in the hyperoxia model. To test these hypotheses, we injected wildtype mice with 0, 10, 20, or 30 mg/kg of 1D11 from P2-P10 in either normoxia or hyperoxia and then harvested at P40 for analysis by histology (*Figure 5A*). To our surprise, we found that while only the highest dose of 1D11 treatment caused alveolar simplification in normoxia, all doses of 1D11 treatment caused worse alveolar simplification in hyperoxia compared to PBS (*Figure 5B*). Consistent with these histology results, lung physiology studies of 1D11-treated mice showed a pattern of emphysematous changes with increased compliance and decreased elastance (*Figure 5C*). These changes are similar to what we previously observed in hyperoxia-treated mice and in *Tgfbr2<sup>F/F</sup>*;Nkx2.1-cre mice and complement the histologic findings of alveolar simplification in 1D11-treated mice. Together, these results support the alternative hypothesis that TGFβ signaling plays a homeostatic role in both normal alveolar development and in hyperoxia.

Because 1D11 antibody treatment is systemic, we were unable to attribute the results of these studies to a specific cell type within the lung. Our current study, consistent with earlier reports, demonstrates that TGFβ signaling to the epithelium is required for normal alveolar development as well as an adaptive response to hyperoxia (*Sureshbabu et al., 2015*). To interrogate the role of TGFβ signaling to the lung mesenchyme, we sought to cross *Tgfbr2<sup>F/F</sup>* mice to either the *Pdgfra-CreERT2* or *Gli1-CreERT2* allele to target *Tgfbr2* in myofibroblasts and mesenchymal cells during alveologen-esis. While *Pdgfra-CreERT2* has been used to specifically target PDGFRα+ cells (*Chung et al., 2018*), many studies have also used the broader *Gli1-CreERT2* to target myofibroblasts during alveolar lung development (*Gao et al., 2022*; *Li et al., 2019*). Because the *Pdgfra-CreERT2* and *Gli1-CreERT2* alleles were generated using a knock-in/knock-out approach, mice carrying the CreERT2 allele are haploinsufficient for *Pdgfra* or *Gli1*, respectively (*Chung et al., 2018*; *Ahn and Joyner, 2004*). We conducted hyperoxia studies in each of these two lines and found that the *Gli1-CreERT2* allele alone does not disrupt alveolar development in either normoxia or with hyperoxia treatment (*Figure 6—figure supplement 1*). In contrast, mice with the *Pdgfra-CreERT2* allele undergo normal development in normoxia, but they develop worse alveolar simplification with hyperoxia treatment compared to their cre-negative littermates (*Figure 6—figure supplement 1*). These data support use of the *Gli1-CreERT2* allele to target the lung mesenchyme in either normoxia or hyperoxia while limiting the use of *Pdgfra-CreERT2* mice (and other *Pdgfra*-haploinsufficient alleles) to normoxic conditions.

Based on these validation studies, we generated *Tgfbr2<sup>F/F</sup>*;*Gli1-CreERT2* mice to study the effects of TGFβ signaling to the lung mesenchyme. Interestingly, we observed a similar pattern of lung disease as 1D11 treatment when deleting *Tgfbr2* in the lung mesenchyme: *Tgfbr2<sup>F/F</sup>*;*Gli1-CreERT2* mice devel-oped alveolar simplification in normoxia when compared to littermate controls, and they developed worse disease than their littermates with hyperoxia exposure (*Figure 6—figure supplement 2A and*

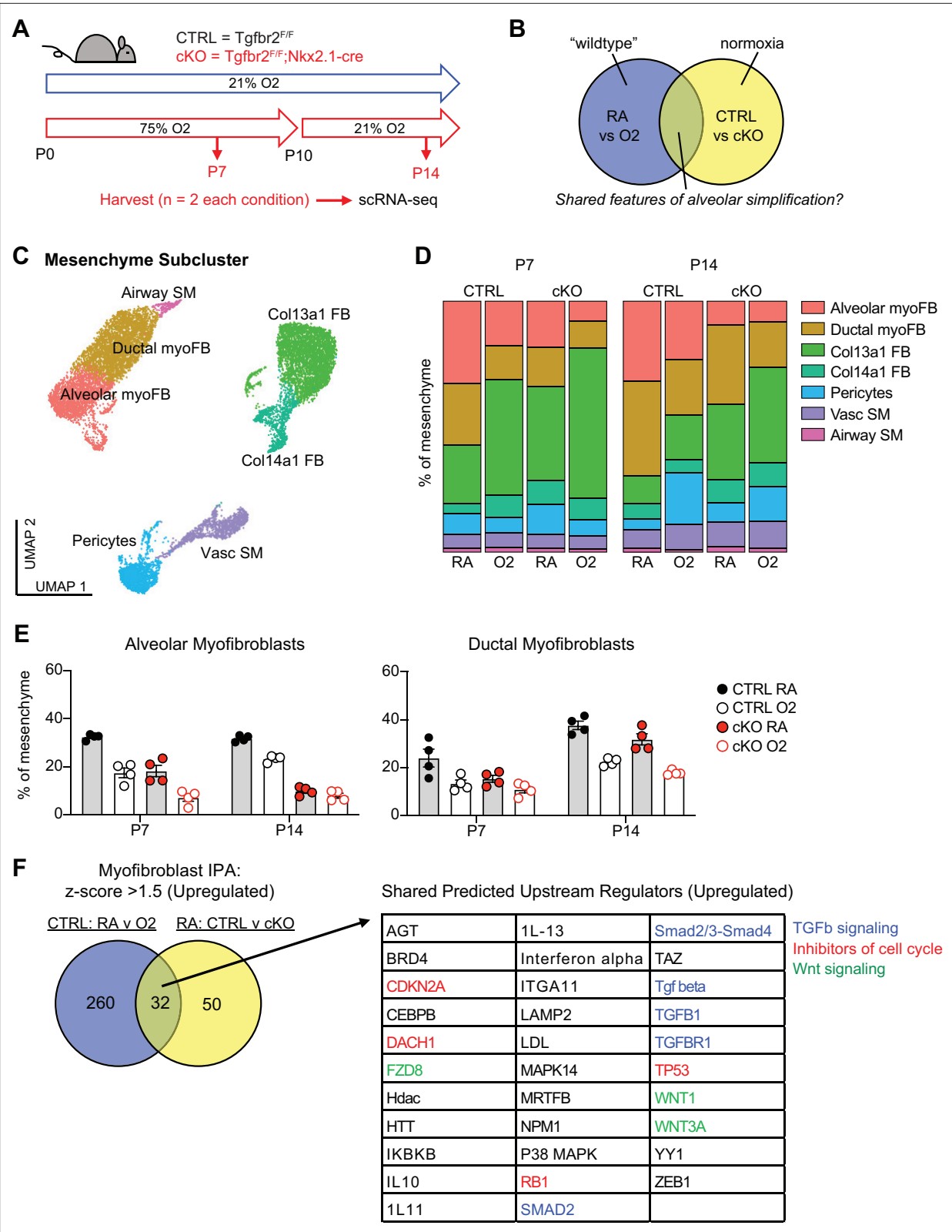

**Figure 3.** scRNA-seq reveals loss of myofibroblasts in both models of alveolar simplification. (**A**) Schematic of scRNA-seq project showing CTRL (*Tgfbr2^{F/F}*) and cKO (*Tgfbr2^{F/F}*;Nkx2.1-cre) littermates treated in 75% hyperoxia versus normoxia from P0-P10 and harvested on P7 or P14 for FACS-purification and analysis by scRNA-seq. n=2 mice harvested for each genotype, treatment condition and timepoint. (**B**) Venn-diagram depicting strategy to compare these two models to identify shared features of alveolar simplification. (**C**) UMAP projection of mesenchymal cells from scRNA-seq as

*Figure 3 continued on next page*

*Figure 3 continued*

outlined in (**A**). (**D**) Bar graphs depicting the frequency of each cell type by treatment condition, genotype, and timepoint. (**E**) The frequency of alveolar and ductal myofibroblasts within the mesenchyme as depicted in (**D**) with each data point representing either a biologic or technical replicate. Because the data depicts both biologic and technical replicates, no statistics were performed on this data set. (**F**) Differentially expressed genes in myofibroblasts were identified by comparing either CTRL RA vs O2 cells or RA CTRL vs cKO cells. These lists were subsequently analyzed by Qiagen IPA to identify predicted upstream regulators for each comparison. The Venn-diagram on left depicts the number of overlapping predicted upstream regulators with z-score >1.5 (upregulated), while the table on right lists these 32 shared upstream regulators. Blue = TGFβ signaling, red = inhibitors of cell cycle, green = Wnt signaling.

The online version of this article includes the following figure supplement(s) for figure 3:

**Figure supplement 1.** scRNA-seq of murine lungs with neonatal hyperoxia treatment or loss of epithelial TGFβ Signaling.

**Figure supplement 2.** Characterization of mesenchymal cell clusters by scRNA-seq.

**Figure supplement 3.** Re-analysis of published data confirms loss of myofibroblasts with neonatal hyperoxia treatment.

*B*). Together, these results show that TGFβ signaling is required to both the lung epithelium and mesenchyme for normal development, that it plays a homeostatic role in response to neonatal hyperoxia treatment, and that all our efforts to inhibit TGFβ made hyperoxia-induced alveolar simplification worse.

To address mechanisms of TGFβ activation in normal alveolar development and in the hyperoxia model of alveolar simplification, we generated *Itgb6^{F/F}* mice and crossed them to the Nkx2.1-cre allele. The αvβ6 integrin is expressed predominantly in epithelial tissues, activates TGFβ in vivo, and has been shown to be a critical mediator of lung injury in models of acute lung injury and pulmonary fibrosis (*Morris et al., 2003*; *Munger et al., 1999*; *Yokosaki et al., 1996*; *Huang et al., 1996*; *Hogmalm et al., 2010*; *Massagué and Sheppard, 2023*). *Itgb6^{F/F}*;Nkx2.1-cre mice lack αvβ6 on lung epithelial cells and undergo normal alveolar development, but to our surprise, show no difference in disease severity with hyperoxia treatment (*Figure 6—figure supplement 2C and D*). Within the mesenchyme, αvβ1 and αvβ8 have been shown to activate TGFβ in vivo (*Massagué and Sheppard, 2023*; *Henderson et al., 2013*; *Reed et al., 2015*; *Mu et al., 2002*), so we next deleted all αv-integrins in the lung mesenchyme by crossing *Itgav^{F/F}* mice to the *Gli1-CreERT2* allele. These mice develop spontaneous alveolar simplification in normoxic conditions, have lung physiology parameters consistent with an emphysematous phenotype, and develop worse lung disease with hyperoxia exposure (*Figure 6A–C*). These data suggest that epithelial αvβ6 does not play a role in TGFβ activation during normal development or neonatal hyperoxia, while αv-integrins in the lung mesenchyme are required for normal development and play a protective role in response to hyperoxia.

## Impaired proliferation of PDGFRα+ fibroblasts with hyperoxia treatment

Flow cytometry and scRNA-seq both showed a reduction of PDGFRα+ fibroblasts in hyperoxia-treated mice, which suggests these cells are exquisitely sensitive to injury. To address whether this might be due to impaired proliferation as suggested by IPA analysis, we exposed wildtype mice to hyperoxia and treated them with the nucleoside analogue 5-ethynyl-2′-deoxyuridine (EdU) to identify proliferating cells. We found a decrease of PDGFRα+ cells in hyperoxia-treated mice by P8 both by percentage and by absolute number (*Figure 7A–C*), which persisted to P10 and continued even with 4 days of normoxic recovery. Using EdU analysis, we found hyperoxia caused a proliferative defect in PDGFRα+ cells as early as P4, preceding the decrease in cell number. This early impairment of proliferation by PDGFRα+ cells was the most striking change observed across all subsets in our EdU analysis, and suggests that a proliferative defect contributes to the decreased cell number.

After observing impaired proliferation of PDGFRα+ cells in the hyperoxia model of lung injury, we explored whether this mechanism is conserved across our other models of alveolar simplification. We used EdU treatment and flow cytometry to characterize cell subsets and proliferation in *Tgfbr2^{F/F}*;Nkx2.1-cre mice and 1D11-treated wildtype mice. Remarkably, both models showed a similar reduction of PDGFRα+ cells along with decreased PDGFRα+ cell proliferation as quantified by EdU uptake (*Figure 8A–D*). These results provide functional validation of our IPA analyses showing enrichment of cell cycle inhibitory pathways in injured myofibroblasts, and they suggest impaired PDGFRα+ cell

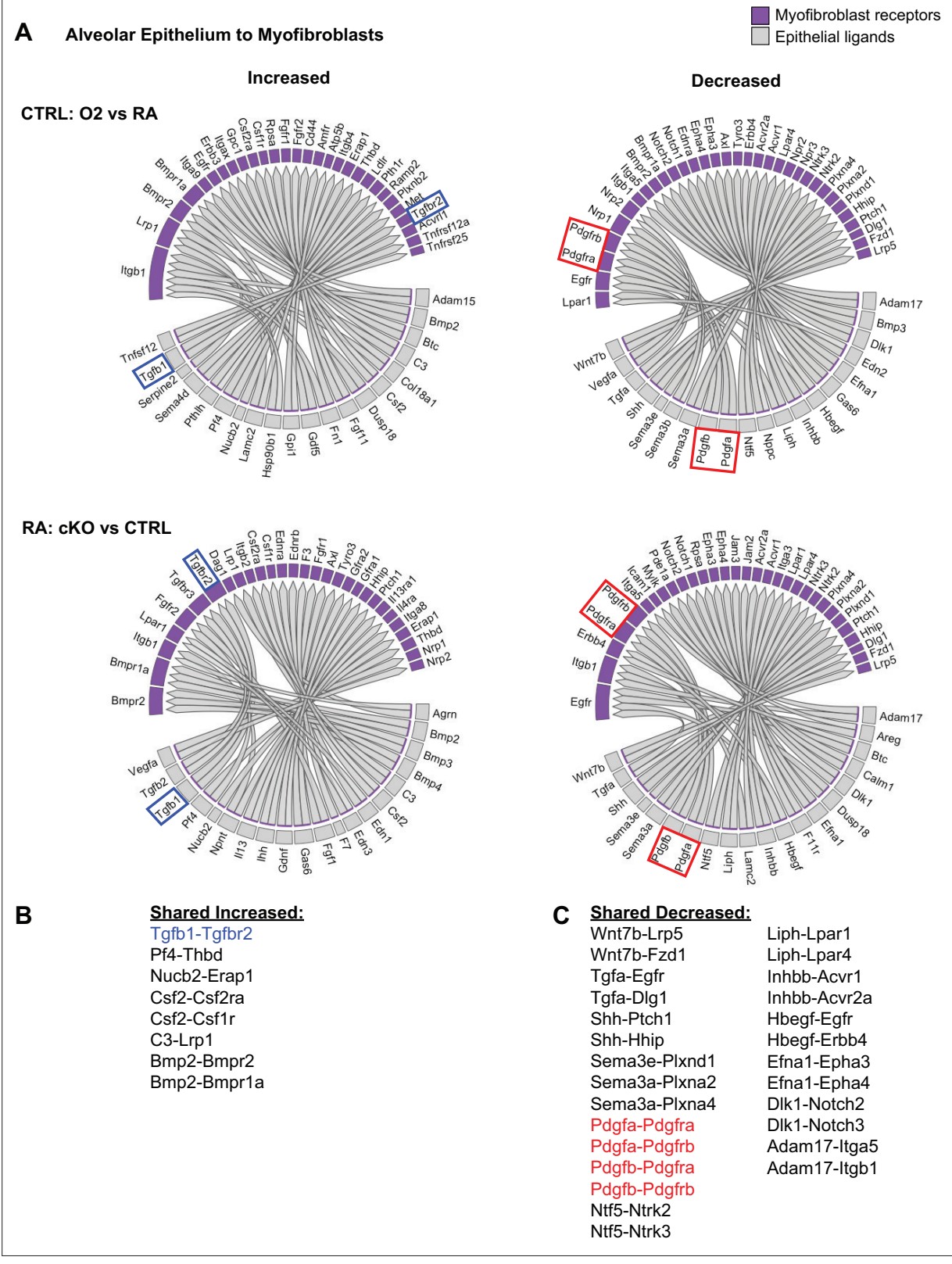

**Figure 4.** NicheNet ligand-receptor analysis of epithelial-mesenchymal crosstalk in both models of alveolar simplification. scRNA-seq data was analyzed using the NicheNet software package. (**A**) Alveolar epithelial clusters were pooled to define sender population and myofibroblast clusters were pooled to define receiver population. By analyzing differential expression of ligands, receptors, and downstream gene expression changes in myofibroblasts, NicheNet predicted increased (left) or decreased (right) ligand-receptor signaling in each comparison of interest. Gray boxes show ligands expressed

*Figure 4 continued*

on the epithelium and the purple-shaded boxes indicate the corresponding receptors expressed on myofibroblasts. Upper panels compare hyperoxia versus normoxia in CTRL cells. Lower panels compare CTRL versus cKO cells in normoxia. (**B**) List of ligand-receptor pairs predicted to be increased in both injury models. Tgfb1-Tgfbr2 pairings highlighted in blue. (**C**) List of ligand-receptor pairs predicted to be decreased in both injury models. Pdgfa-Pdgfra and Pdgfb-Pdgfrb pairings highlighted in red.

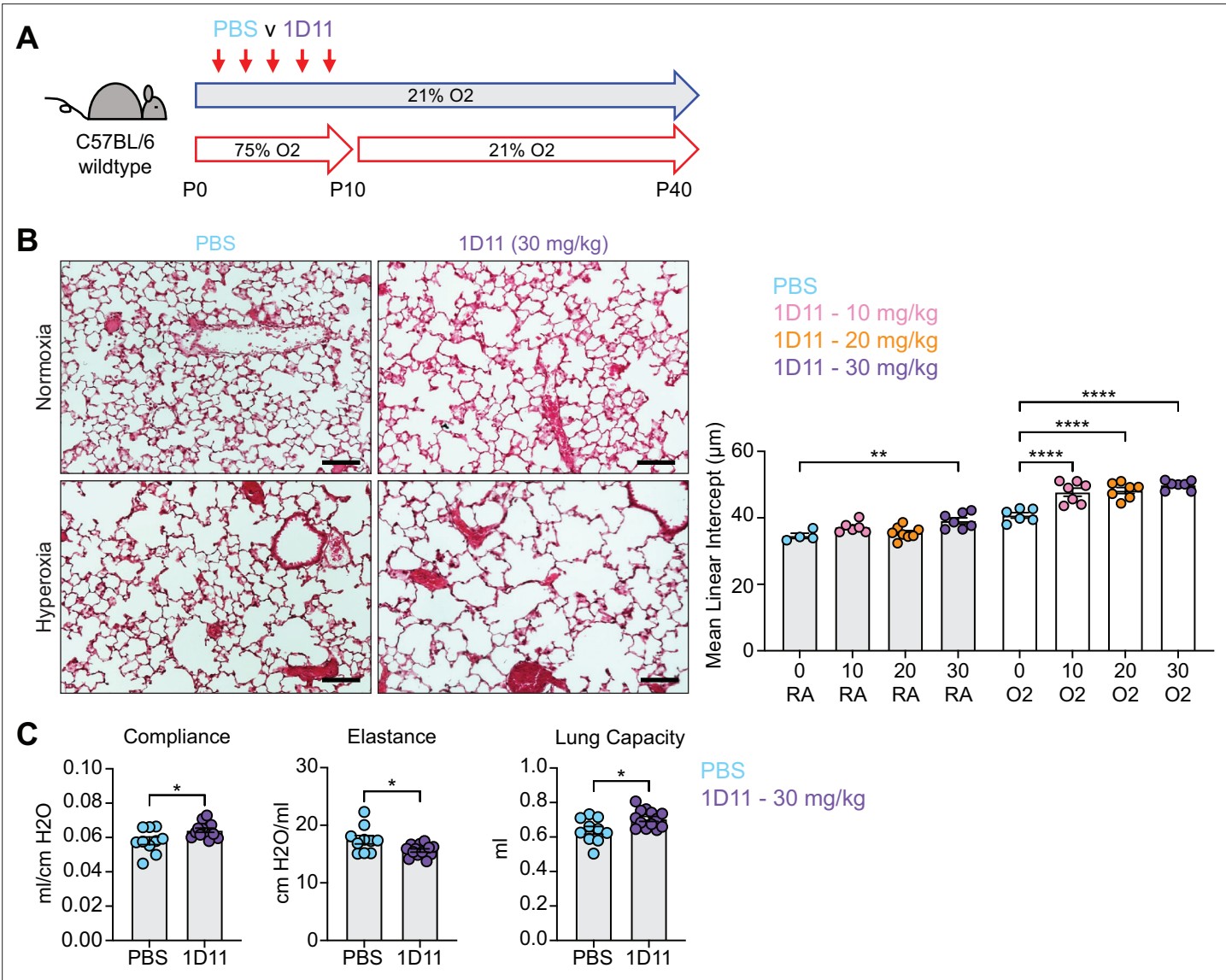

**Figure 5.** Inhibiting TGFβ disrupts alveolar development and exacerbates hyperoxia-induced injury. (**A**) Wildtype C57BL/6 mice were injected every other day from P2-P10 with PBS or 1D11 (pan- TGFβ-blocking antibody), treated in 75% hyperoxia treatment versus normoxia from P0-P10, and recovered in room air until harvest at P40 for analysis by either histology or lung physiology. (**B**) H&E sections of representative lungs from (**A**) harvested at P40 (left). Images shown are from PBS and 30 mg/kg 1D11 treatment groups. Mean linear intercepts calculated for all treatment groups (right). (**C**) PBS- and 30 mg/kg 1D11-treated mice treated in normoxic conditions as in (**A**) and harvested for lung physiology measurements of compliance, elastance and lung capacity at P40. Data in (**B**) compared by ANOVA with Fisher's post hoc test; for readability and limitations of graphing, only the statistical significance values within normoxia or hyperoxia cohorts are plotted. Data in (**C**) compared by two-tailed unpaired Student's t-test. Error bars depict mean ± SEM. *p<0.05, **p<0.01, ****p<0.0001. Scale bars = 100 μm.

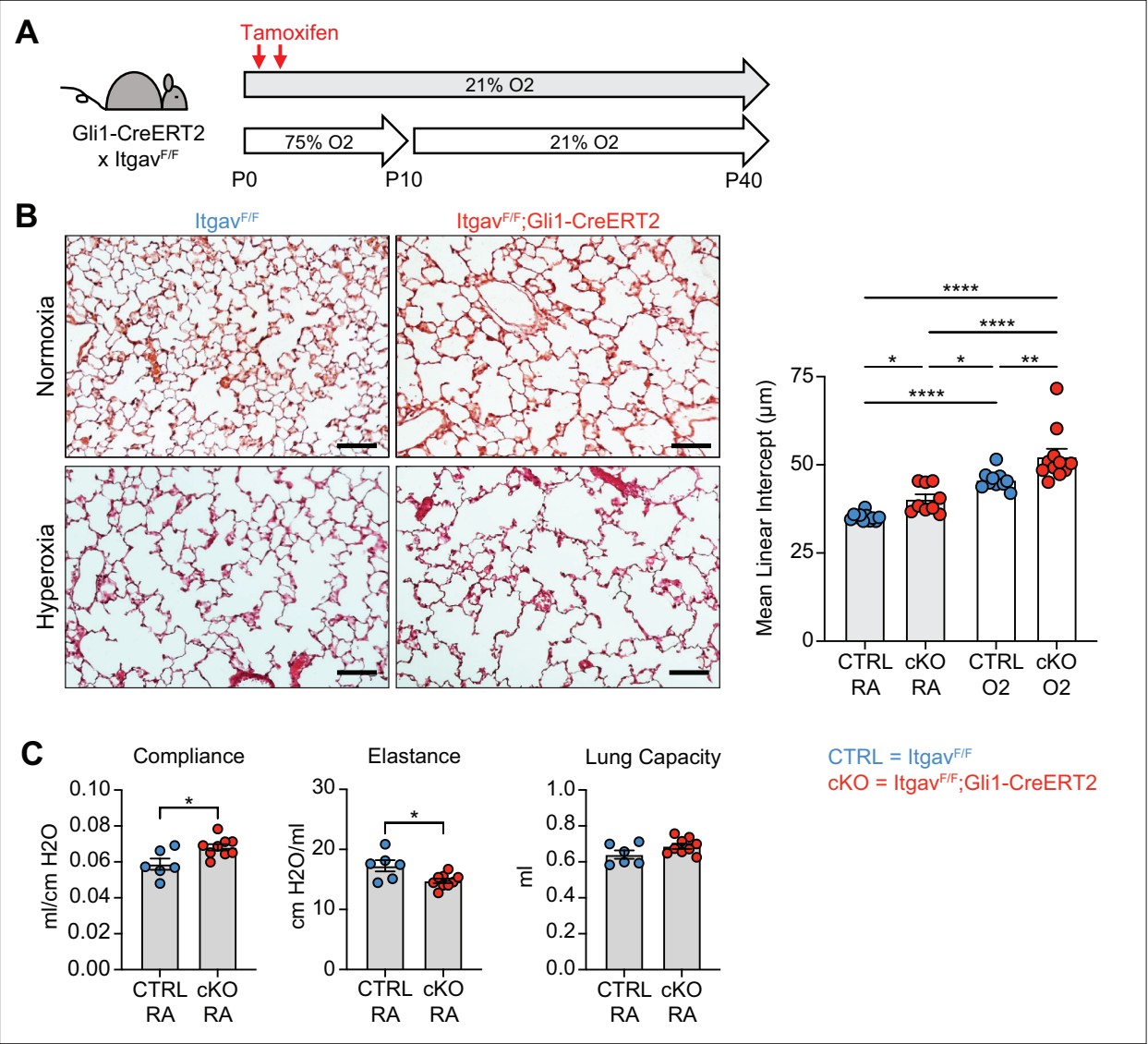

**Figure 6.** Deletion of αv-integrins in lung mesenchyme impairs alveolar development and worsens hyperoxia-induced injury. (**A**) *Itgav^{F/F}* and *Itgav^{F/F};Gli1-CreERT2* littermates were injected with tamoxifen on P2 and P4, treated in 75% hyperoxia versus normoxia from P0-P10, and recovered in room air until harvest at P40 for analysis by either histology or lung physiology. (**B**) H&E sections of representative lungs from (**A**) harvested at P40 (left). Mean linear intercepts calculated for all treatment groups (right). (**C**) Normoxia cohort treated as in (**A**) and harvested for lung physiology measurements of compliance, elastance and lung capacity. Data in (**B**) compared by ANOVA with Fisher's post hoc test. Data in (**C**) compared by two-tailed unpaired Student's t-test. Error bars depict mean ± SEM. *p<0.05, **p<0.01, ****p<0.0001. Scale bars = 100 μm.

The online version of this article includes the following figure supplement(s) for figure 6:

**Figure supplement 1.** *Gli1-CreERT2* allele does not disrupt alveolar development, but *Pdgfra-CreERT2* allele worsens hyperoxia-induced injury.

**Figure supplement 2.** Loss of TGFβ signaling to lung mesenchyme causes worse disease in hyperoxia while *Itgb6* plays no role in alveolar development.

proliferation might be a conserved feature of developmental lung injuries which result in alveolar simplification.

## Impaired proliferation of PDGFRα+ fibroblasts is sufficient to cause alveolar simplification

After observing impaired PDGFRα+ cell proliferation in multiple models of alveolar simplification, we sought to determine whether inhibiting proliferation in these cells would be sufficient to cause disease. We generated mice with conditional deletion of *Ect2*, a protein required for cytokinesis (*Cook et al.,*

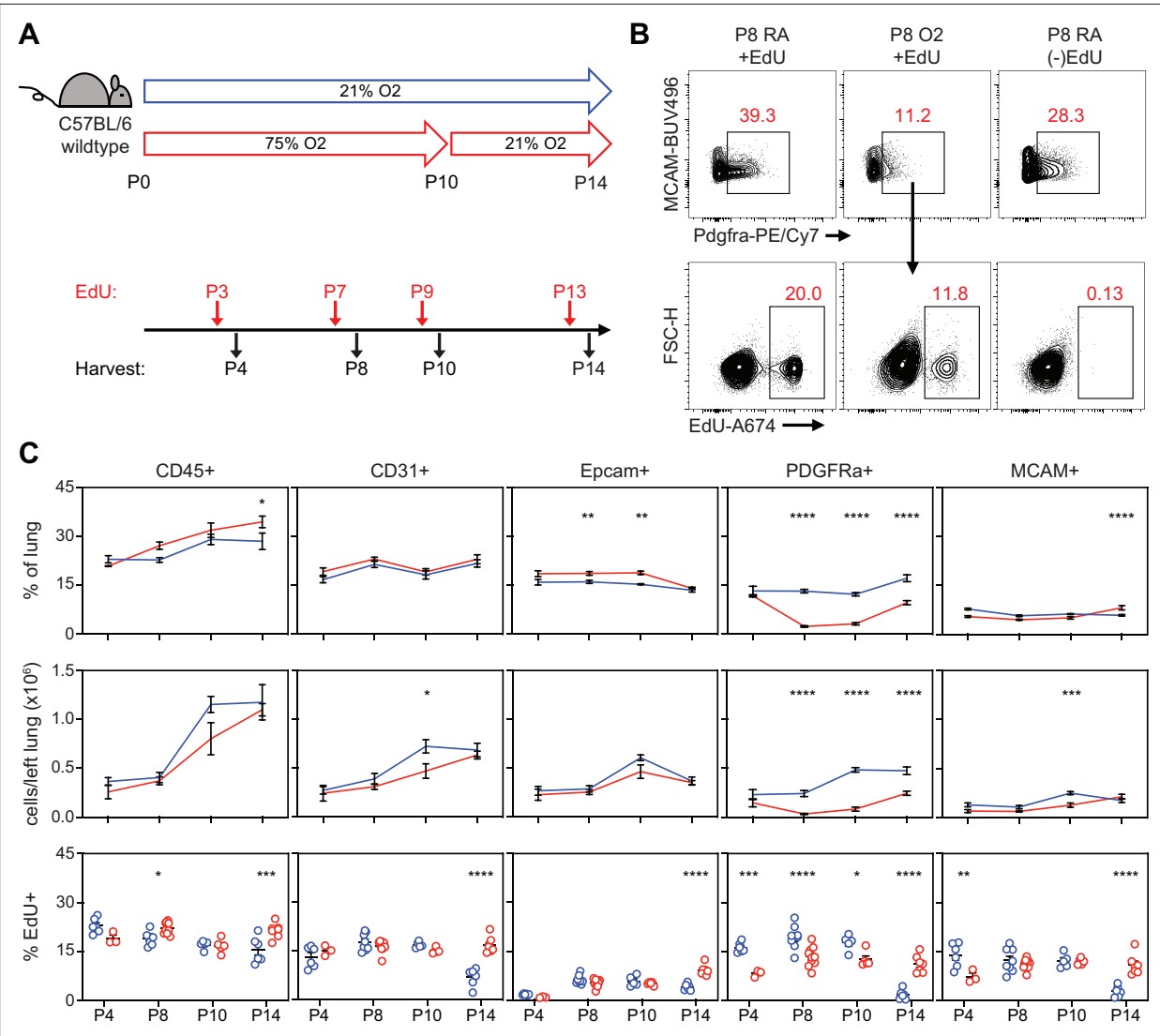

**Figure 7.** Impaired proliferation of PDGFRα+ fibroblasts with neonatal hyperoxia treatment. (**A**) Wildtype C57BL/6 mice were treated in 75% hyperoxia versus normoxia from P0-P10 and recovered in room air until indicated timepoints for analysis. Mice were injected with EdU 24 hr prior to each analysis timepoint. (**B**) Flow cytometry plots of lung mesenchyme (CD45-, CD31-, Epcam-, MCAM-) show gating of PDGFRα+ cells (upper panels) and subsequent identification of EdU+/PDGFRα+ cells (lower panels). Panels on far-right show an EdU-untreated littermate used to define EdU + gate. (**C**) Time course graphs showing indicated populations of the lung as percent of lung (top), total cells in left lung (middle), and percent EdU + cells (bottom). Data graphed as mean ± with exception of EdU + panel in which each animal is plotted individually. Data compared by two-tailed unpaired Student's t-test. *p<0.05, **p<0.01, ***p<0.001, ****p<0.0001.

*2011*), by crossing the *Ect2*F/F mice with the *Pdgfra-CreERT2* allele. We found earlier that PDGFRα+ cells underwent robust proliferation through P10 in normoxic conditions (***Figure 7C***), so we analyzed *Ect2*F/F;*Pdgfra-CreERT2* mice at P14 to characterize the impact of this mouse model on the cellular composition of the lung during early alveologenesis. Using flow cytometry, we observed a significant reduction in PDGFRα+ cells within the lung by percentage and absolute number (***Figure 9A and B***). While this reduction in PDGFRα+ cells was expected in *Ect2*F/F;*Pdgfra-CreERT2* mice, it was interesting that *Pdgfra* expression was reduced in the remaining PDGFRα+ cells as quantified by mean fluorescence intensity (MFI). Earlier studies found a correlation between *Pdgfra* expression levels and fibroblast proliferation (***Kimani et al., 2009***), so we wondered whether this phenotype of reduced *Pdgfra* expression was conserved across the other injury models in this study which showed reduced PDGFRα+ cell proliferation. Indeed, quantification of PDGFRα MFI by flow cytometry confirmed that

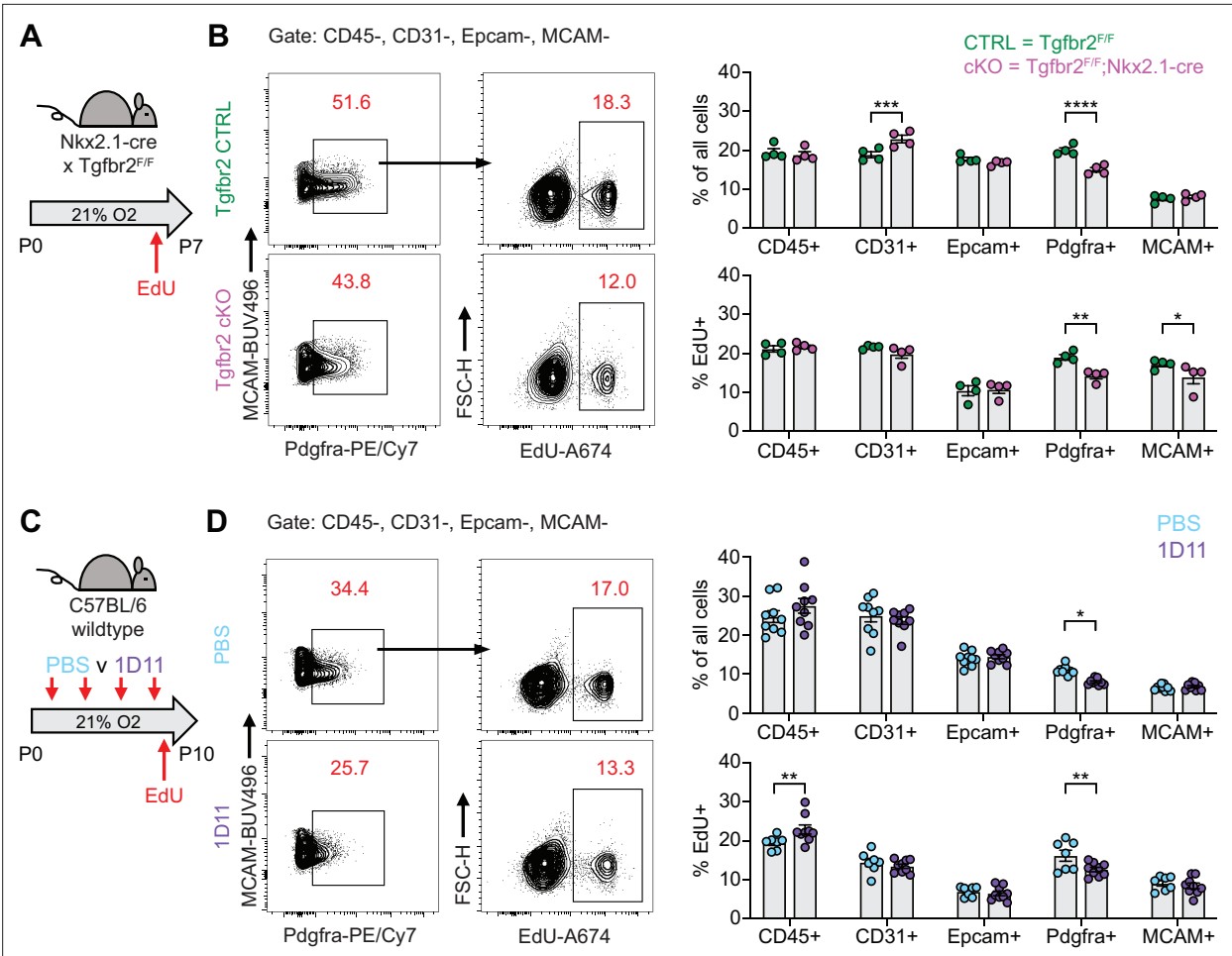

**Figure 8.** Impaired PDGFRα+ cell proliferation is a conserved feature across multiple models of alveolar simplification. (**A**) *Tgfb2^{F/F}* and *Tgfbr2^{F/F}*;Nkx2.1-cre littermates were maintained in normoxic conditions from P0-P7, injected with EdU on P6, and harvested 24 hr later on P7 for flow cytometry. (**B**) Flow cytometry plots of lung mesenchyme (CD45-, CD31-, Epcam-, MCAM-) show gating of PDGFRα+ cells (left panels) and subsequent identification of EdU+/PDGFRα+ cells (right panels). Graphs on far right show major cell populations of the lung by percentage (upper graphs) and percent EdU-positive within each of these populations (lower graphs). (**C**) Wildtype C57BL/6 mice were injected every other day from P2-P8 with PBS or 30 mg/kg 1D11 (pan-TGFβ-blocking antibody) in normoxic conditions, injected with EdU on P9, and harvested 24 hr later on P10 for flow cytometry. (**D**) Flow cytometry plots of lung mesenchyme (CD45-, CD31-, Epcam-, MCAM-) show gating of PDGFRα+ cells (left panels) and subsequent identification of EdU+/PDGFRα+ cells (right panels). Graphs on far right show major cell populations of the lung by percentage (upper graphs) and percent EdU-positive within each of these populations (lower graphs). Data in analyzed by two-tailed unpaired Student's t-test. Error bars depict mean ± SEM. *p<0.05, **p<0.01, ***p<0.001, ****p<0.0001.

*Pdgfra* expression is significantly reduced in the lung mesenchyme and specifically in PDGFRα+ cells in each of the injury models we analyzed in this study (*Figure 9—figure supplement 1*).

Next, we aged *Ect2^{F/F}*;*Pdgfra-CreERT2* mice until P40 to analyze their lungs by either morphometry or lung physiology. By histology, we observed enlarged alveolar airspaces and increased MLI's in conditional knockouts (*Ect2^{F/F}*;*Pdgfra-CreERT2*) compared to controls (*Figure 9C*). Importantly, conditional heterozygous mice showed no difference in MLI compared to cre-negative mice, which confirms that haploinsufficiency of *Pdgfra* caused by using the *Pdgfra-CreERT2* allele in normoxic conditions is not responsible for this phenotype (*Figure 9C* and *Figure 6—figure supplement 1*). By lung physiology, *Ect2^{F/F}*;*Pdgfra-CreERT2* mice produced the same emphysematous phenotype of increased compliance and decreased elastance which we observed in several other models of alveolar simplification in this study (*Figure 9D*). Together, these data demonstrate that impaired proliferation of PDGFRα+ cells is sufficient to cause alveolar simplification, even in the absence of environmental insult or modulation of TGFβ signaling.

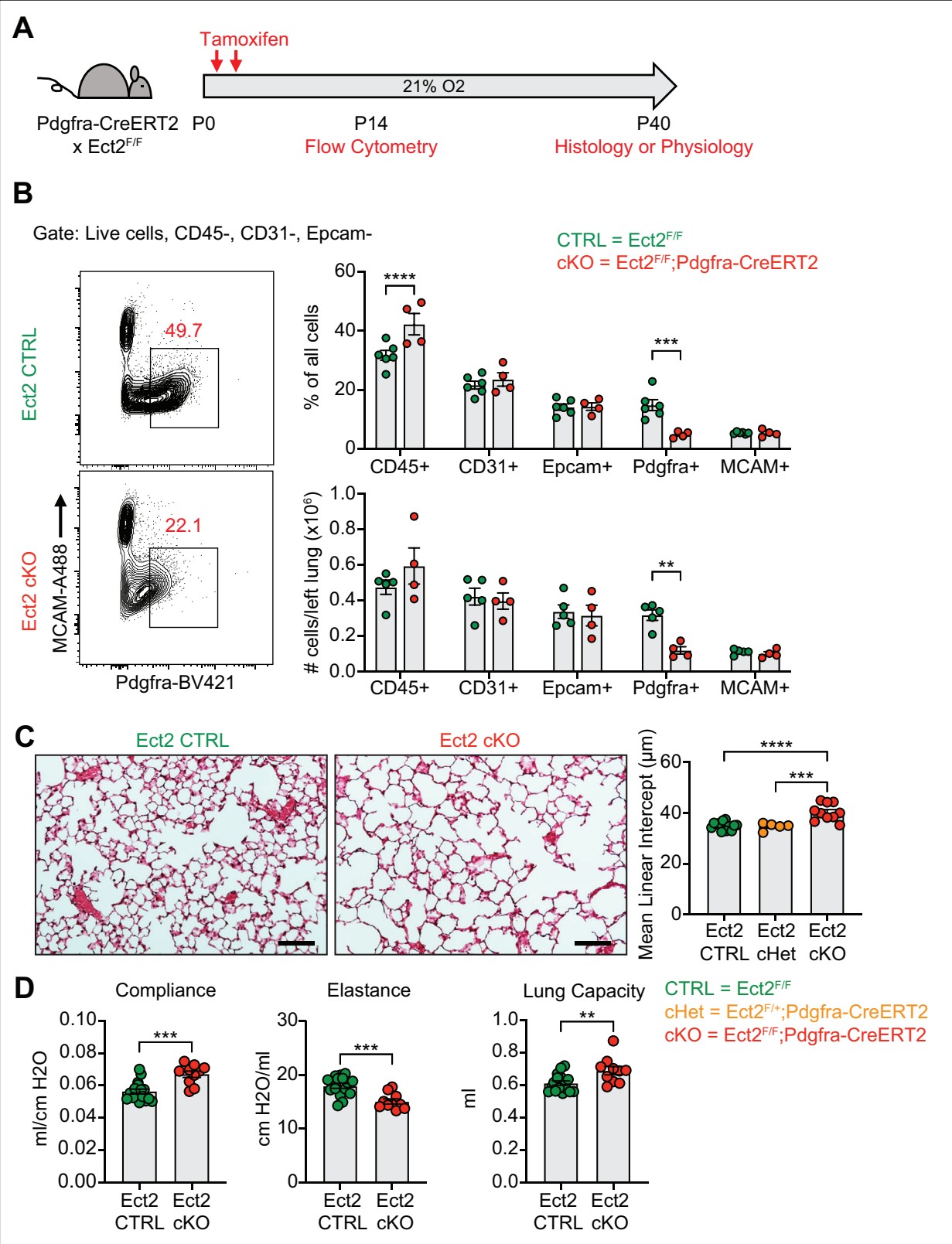

**Figure 9.** Blocking proliferation of PDGFRα+ fibroblasts is sufficient to cause alveolar simplification. (**A**) *Ect2^F/F* and *Ect2^F/F;Pdgfra-CreERT2* littermates were injected with tamoxifen on P2 and P4 in normoxic conditions. Mice were analyzed by flow cytometry on P14 or aged until P40 for analysis by either histology or lung physiology. (**B**) Representative flow cytometry plots of the lung mesenchyme (live, CD45-, CD31-, and Epcam-) with gates depicting PDGFRα+ cells (left). Major cell populations of the lung were defined by the indicated cell surface markers and shown as either a percentage of all cells

*Figure 9 continued on next page*

*Figure 9 continued*

(top) or as absolute number (bottom). (**C**) H&E sections of representative lungs from (**A**) harvested at P40 (left). Mean linear intercepts calculated for all treatment groups (right). (**D**) Mice treated as in (**A**) and harvested for lung physiology measurements of compliance, elastance and lung capacity. Data in (**B**) compared by two-tailed unpaired Student's t-test. Data in (**C**) compared by ANOVA with Fisher's post hoc test. Error bars depict mean ± SEM. **p<0.01, ***p<0.001, ****p<0.0001. Scale bars = 100 μm.

The online version of this article includes the following figure supplement(s) for figure 9:

**Figure supplement 1.** Decreased PDGFRα mean fluorescence intensity across multiple models of alveolar simplification.

## Discussion

Using a combination of flow cytometry, lung physiology, and scRNA-seq, we identified several core features of lung injury that are conserved across multiple mouse models of alveolar simplification. We first compared neonatal hyperoxia exposure (75% hyperoxia P0-P10) to the loss of epithelial TGFβ signaling (*Tgfbr2*$^{F/F}$;Nkx2.1-cre). Flow cytometric analysis of both models showed a significant reduction of PDGFRα+ cells, while scRNA-seq studies attributed these changes to a loss of alveolar and ductal myofibroblasts. Qiagen IPA and NicheNet ligand-receptor analyses identified several pathways that were upregulated in injured myofibroblasts in both models: increased TGFβ signaling, enrichment of inhibitors of cell cycle, and decreased Pdgfa-Pdgfra signaling. Using a combination of pharmacologic and genetic approaches, we demonstrated that increased TGFβ signaling in the mesenchyme does not seem to be a driver of alveolar simplification, but rather TGFβ signaling is critical for both normal alveolar development and for protecting against impaired alveolar development in response to hyperoxia. In contrast, we show that PDGFRα+ cell proliferation is reduced in multiple models of alveolar simplification, and that this impaired proliferation on its own is sufficient to cause alveolar simplification.

Several recent studies have identified lung mesenchymal cells as critical mediators of alveologenesis. Li et al. established the requirement of myofibroblasts during postnatal development by specifically ablating PDGFRα+ cells at the onset of alveologenesis and showing that these mice developed alveolar simplification (*Li et al., 2018*). Other work has identified essential functions of myofibroblasts during alveolar development, including contractility (*Li et al., 2020*) and mitochondrial energetics (*Zhang et al., 2022*). Ricetti et al used bulk RNA sequencing to show a skewing from myofibroblast to matrix fibroblast phenotypes after hyperoxia exposure along with reduced Ki-67 uptake and contractility in myofibroblasts (*Riccetti et al., 2022*). Our current study builds on this existing body of work by characterizing significant changes in PDGFRα+ cell number in several models of developmental lung injury. By flow cytometry, we found reduction in the number of PDGFRα+ cells in hyperoxia-treated mice and in *Tgfbr2*$^{F/F}$;Nkx2.1-cre mice during the early phase of alveologenesis. Using scRNA-seq, we attributed this reduction to the alveolar and ductal myofibroblast populations. Importantly, we identified myofibroblast proliferation as a critical feature of normal alveolar development and validated this observation in several experimental models.

NicheNet analysis of our scRNA-seq data predicted decreased Pdgfa-Pdgfra signaling from the epithelium to the myofibroblasts in both hyperoxia-treated and *Tgfbr2*$^{F/F}$;Nkx2.1-cre mice. This observation is interesting because several studies have focused on the requirement of Pdgfa-Pdgfra interactions for myofibroblast identity and function. Deletion of *Pdgfra* in mesenchymal populations of the neonatal lung has been shown to disrupt alveologenesis (*Li et al., 2019*; *McGowan and McCoy, 2014*). Other studies have found lower *Pdgfra* expression in lung samples from BPD patients (*Popova et al., 2014*; *Oak et al., 2017*). In our current study, we use flow cytometry to show a loss of PDGFRα+ cells in multiple models of alveolar simplification: hyperoxia-treated mice, *Tgfbr2*$^{F/F}$;Nkx2.1-cre mice, 1D11-treated mice, and Ect2$^{F/F}$;Pdgfra-CreERT2 mice. In addition to the loss of PDGFRα+ cells, we also observed decreased *Pdgfra* expression within the remaining PDGFRα+ cells (*Figure 9—figure supplement 1*). Given that EdU uptake is also reduced amongst the remaining PDGFRα+ cells, these results suggest Pdgfa-Pdgfra signaling might regulate myofibroblast proliferation in addition to their identity and function during neonatal lung development.

As part of our current studies, we conducted control experiments with the *Gli1-CreERT2* and *Pdgfra-CreERT2* alleles and discovered results with broad implications for others utilizing similar genetic tools for their work. Both alleles were generated using a knock-in/knock-out gene targeting approach which disrupts the native expression of *Gli1* and *Pdgfra*, respectively (*Chung et al., 2018*; *Ahn and Joyner,*

*2004*). Because Hedgehog and Pdgfa-Pdgfra signaling pathways are important for lung development (*Kugler et al., 2017*; *Yie et al., 2023*), we sought to validate the effects of haploinsufficiency of *Gli1* and *Pdgfra* when using these alleles. Both *Gli1-CreERT2* and *Pdgfra-CreERT2* mice underwent normal alveolar development in normoxic conditions when compared to cre-negative littermates. In contrast, *Pdgfra-CreERT2* mice developed worse alveolar simplification after hyperoxia treatment compared to their cre-negative littermates. The *Gli1-CreERT2* results suggest this allele can be used for conditional knockout studies in either normoxic or hyperoxic conditions with little impact from *Gli1* haploinsufficiency. The *Pdgfra-CreERT2* results, however, should serve as a caution to researchers utilizing genetic tools with *Pdgfra*-haploinsufficiency.

Recent work has established an essential role for TGFβ signaling in normal alveolar development (*Calthorpe et al., 2023*), but the functional role of TGFβ in models of BPD and alveolar simplification remains unclear. Our scRNA-seq data showed increased TGFβ signaling in myofibroblasts in two injury models, so we hypothesized that excess TGFβ signaling might be a driver of disease pathogenesis. To determine the functional significance of TGFβ in the hyperoxia model, we treated mice with the pan-TGFβ-blocking antibody 1D11. Because TGFβ is required for normal alveologenesis, we hypothesized a dose titration approach might identify a 'sweet spot' to neutralize an excess of TGFβ in hyperoxia while preserving normal development in normoxia. Instead, we found that every dose of 1D11 treatment worsened hyperoxia-induced alveolar simplification while only the highest dose caused disease in normoxia. While the 1D11-normoxia results confirm an essential role for TGFβ in normal alveolar development, the 1D11-hyperoxia results suggest TGFβ plays a homeostatic function rather than a pathologic role in the hyperoxia model of injury. Because 1D11 treatment neutralizes TGFβ ligands systemically, we used Nkx2.1-cre and *Gli1-CreERT2* alleles to conditionally delete *Tgfbr2* in the lung epithelium and mesenchyme, respectively. Confirming earlier reports, we found that disrupting TGFβ signaling to either of these populations causes alveolar simplification (*Sureshbabu et al., 2015*; *Gao et al., 2022*). Of significance, however, both *Tgfbr2*<sup>F/F</sup>;Nkx2.1-cre and *Tgfbr2*<sup>F/F</sup>;Gli1-CreERT2 mice developed worse disease with neonatal hyperoxia treatment. Taken together with our 1D11-hyperoxia experiments, we conclude that TGFβ signaling to both the lung epithelium and mesenchyme is required for normal alveolar development and is protective rather than pathologic in hyperoxia-perturbed alveolar development.

Little is known about mechanisms of TGFβ activation in alveolar development and neonatal lung injury. TGFβ ligands are secreted as latent, inactive complexes that are anchored to the cell surface or the extracellular matrix (*Massagué and Sheppard, 2023*). These complexes prevent TGFβ from engaging its receptors, and therefore TGFβ activation is tightly regulated in tissues despite the high levels of latent TGFβ complex (*Massagué and Sheppard, 2023*). Integrins are cell-surface proteins that regulate interactions in the extracellular matrix, and several studies by our group have demonstrated that the epithelial integrin αvβ6 is critical for TGFβ activation in models of acute lung injury and pulmonary fibrosis (*Munger et al., 1999*; *Huang et al., 1996*; *Pittet et al., 2001*; *Breuss et al., 1993*). However, TGFβ activation by epithelial αvβ6 does not play a role in alveolar development in either normoxia or hyperoxia. We then deleted αv-integrins in the lung mesenchyme since αvβ1 and αvβ8 are the only other integrins known to activate TGFβ in vivo (*Massagué and Sheppard, 2023*). Interestingly, deletion of αv-integrins under normoxic conditions only produced a modest degree of alveolar simplification but clearly led to more severe alveolar simplification after hyperoxia.

We originally hypothesized that increased TGFβ signaling was a pathologic response to neonatal hyperoxia treatment and that targeting the TGFβ pathway might protect from disease pathogenesis. While our results disproved this hypothesis, they also seem at odds with published data where blocking TGFβ signaling was beneficial in various lung injury models. For example, it was unexpected that *Tgfbr2*<sup>F/F</sup>;Nkx2.1-cre mice develop worse lung injury in our hyperoxia-induced model of BPD because earlier work showed these same mice to be protected from hyperoxia (*Sureshbabu et al., 2015*). We suspect this difference in outcome can be explained by the more aggressive models of lung injury used by this earlier study. Specifically, these authors used either transgenic overexpression of TGFβ or 100% hyperoxia exposure which both resulted in neonatal mortality and severe inflammation, neither of which are observed in our model of 75% hyperoxia. Additionally, many earlier studies focused on TGFβ activation in hyperoxia-induced BPD utilized chronic 85% hyperoxia such as P0-P14 or P0-P28, both of which result in inflammation and fibrosis (*Alejandre-Alcázar et al., 2007*; *Mižíková et al., 2015*; *Witsch et al., 2014a*; *Witsch et al., 2014b*; *Kumarasamy et al., 2009*). Similarly, one

earlier study used transgenic expression of Il-1b in the developing airways to produce a BPD-like phenotype due to inflammation (*Bry et al., 2007*). Deletion of *Itgb6*, and therefore loss of epithelial αvβ6 integrins, conferred partial protection from disease in this injury model (*Hogmalm et al., 2010*). A common theme across these earlier studies implicating TGFβ in pathology is that they utilized aggressive models of lung injury which caused severe inflammation, fibrosis, and neonatal mortality, all of which recapitulate the histopathologic features of 'old BPD' in the clinical setting. In contrast, the hyperoxia model used in our current study more closely represents the 'new BPD' phenotype of emphysematous changes with simplified alveolar structures in the absence of lung scarring and fibrosis (*Jobe, 1999*; *Thébaud et al., 2019*). Collectively, these data suggests that while inhibition of TGFβ may have some protective role in severe injury models associated with significant fibrosis, the major functions of TGFβ signaling during normal alveolar development and in response to moderate hyperoxia are homeostatic and protective.

There are several limitations to our current study. We used flow cytometry to phenotype population-level changes across several models of neonatal lung injury. While flow cytometry is a powerful tool to quantify population subsets and sort cells for scRNA-seq, it requires tissue digestion and generation of single-cell suspensions for downstream analysis. The cells which survive these protocols may represent a skewed population and may not be completely representative of in vivo conditions. For example, many fragile and dying/apoptotic cells do not survive this processing and would be excluded from analysis. Another drawback to flow cytometry is that our ability to define population subsets is limited by the knowledge and feasibility of using cell-surface markers to faithfully distinguish cell types within each population. Recent work by our group has identified cell-surface markers to characterize fibroblast subsets in the adult lung by flow cytometry, but we are unaware of similar protocols to address fibroblast heterogeneity in the neonatal lung (*Tsukui et al., 2020*). We used scRNA-seq to characterize cell subsets within the lung mesenchyme, but the identification and validation of cell-surface proteins suitable for flow cytometric analysis of these same populations is beyond the scope of this current study. Our scRNA-seq data, together with emerging scRNA-seq from multiple other laboratories should provide a rich dataset for others to identify cell surface markers that allow more precise analysis and sorting of distinct mesenchymal populations in the developing lung.

Our current work highlights the essential role of myofibroblast proliferation and TGFβ signaling in normal alveologenesis. We show that myofibroblasts are depleted during neonatal lung injury and that their loss is at least partially due to impaired proliferation and expansion during this critical window of development. We confirm the results of others who similarly observed increased TGFβ signatures in the hyperoxia-injured neonatal lung and have generated interest in targeting this pathway as a therapeutic intervention for BPD. However, after conducting exhaustive studies targeting TGFβ ligands, receptors and activating integrins, we conclude that increased TGFβ signaling in myofibroblasts more likely represents a failed compensatory mechanism rather than a central driver of disease pathogenesis. Clinical BPD remains a heterogeneous disease encompassing both the severe inflammation, scarring and fibrosis of the 'old BPD' phenotype as well as the increasingly prevalent 'new BPD' phenotype of alveolar simplification and emphysematous changes (*Jobe, 1999*; *Thébaud et al., 2019*). Therefore, further work is necessary to contextualize our results with respect to these different clinical BPD phenotypes.

In summary, our results underscore the importance of impaired myofibroblast proliferation as a central feature of alveolar simplification in several models of murine lung injury. Further work is needed to validate these findings in human BPD. If validated, these results would suggest that efforts to prevent or reverse this process could have therapeutic value in BPD. Our sequencing data should provide useful insights to the broader community studying alveolar development and neonatal lung injury. While our current work focuses on a few of the genes and pathways highlighted by these data, we are optimistic that others will utilize this dataset to expand our understanding of the molecular mechanisms driving both normal and aberrant alveolar development.

## Materials and methods
### Mice
C57BL/6 (Stock No. 000664), Nkx2.1-cre (Stock No. 008661), *Pdgfra-CreERT2* (Stock No. 032770), and *Gli1-CreERT2* (Stock No. 007913), and *Itgav^{F/F}* (Stock No. 032297) mouse lines were obtained from

the Jackson Laboratory. *Tgfbr2^{F/F}* mice (exon 2 conditional allele) were described previously (*Chytil et al., 2002*). *Ect2^{F/F}* were described previously (*Cook et al., 2011*), and were obtained from Dr. Alan Fields at Mayo Clinic. All lines were maintained on the C57BL/6 genetic background except for the *Ect2^{F/F}* line, which was originally generated in BALB/c and backcrossed 5 generations to C57BL/6 for these studies. All animal experiments were in accordance with protocols approved by the Institutional Animal Care and Use Committee and Laboratory Animal Resource Center.

## Neonatal hyperoxia treatment

The hyperoxia animal chamber (BioSpherix) was attached to a medical oxygen source controlled by a ProOx single gas controller (BioSpherix) set to maintain 75% oxygen under normobaric conditions. Birth was defined as <12 hr of life, and pups assigned to hyperoxia were transferred into the chamber with lactating dams and maintained from P0-P10 and then recovered in normoxia. During hyperoxia treatment, lactating dams were rotated between hyperoxia and normoxia to prevent maternal injury and to control for nutrition amongst both cohorts. Mice in both conditions were given nestlets and trail mix for additional enrichment. Mice in both conditions remained under typical 7 a-7p light cycling, and chamber was checked daily to monitor temperature, humidity, and gas controller function.

## Tamoxifen, antibody, and EdU treatments

Tamoxifen (Sigma) was dissolved in corn oil (Sigma) at 15 mg/ml, and 150 μg (10 μl) was administered via intraperitoneal (i.p.) injections on P2 and P4. Antibody clone 1D11 was used for pan-TGFβ-blocking studies (*Dasch et al., 1989*), and was generated in our laboratory from a hybridoma obtained from ATCC. Antibody was diluted in PBS and administered via i.p. injections at 0, 10, 20, or 30 mg/kg on P2, P4, P6, P8, and P10. For proliferation studies, EdU (Thermo Fisher) was reconstituted in DMSO at 100 mg/ml, diluted in PBS to 5 mg/ml, and injected i.p. at 75 mg/kg 24 hr prior to harvest.

## Generation of Itgb6 flox mice

*Itgb6* flox mice (*Itgb6^{F/F}*) were made by CRISPR/Cas9-aided homology-directed repair. Loxp sequences were inserted to flank exon 4 of *Itgb6*. Two guide RNA target sequences were chosen in the introns upstream and downstream of exon 4. crRNAs were obtained from IDT with input sequences CAGC TTATCATCCATCTAAA (upstream) and ACCTTCCTCTGACGCACTTT (downstream). Two 200 bp donor DNAs were obtained from IDT. EcoRV sites (gatatc) were inserted following loxp sequences for screening purposes. The sequences of the donor DNAs are as follows:

Upstream donor DNA: TAATCTCTCCTTTATTTGGCTCACCTTTTCTGCAACCACACACCAAGA AAGGGCAGCTTATCATCCATCTAAAATAACTTCGTATAGCATACATTATACGAAGTTATgatatcTGGA TGCTACTTCTCCCTAGGAGATATAAAATATCCCAACATACACCTCCTTCTGTCCTTCAATCCTCAC.

Downstream donor DNA: TAACCTACATTTTTTTCTCTGAGTTTTTCTATCAAAATAACAATTTTTGCACCTTCCTCTGACGCACT TTATAACTTCGTATAGCATACATTATACGAAGTTATgatatcGGGAAATGTGGCTTTCACTCATTGCTGA GAGCAGCAGCCTTCATTGCAATTAAAGTCAAGAGGAAATGGG.

CRISPR/Cas9 complex (Cas9, crRNA, trRNA) and donor DNA were injected into C57BL/6 fertilized zygotes, which were then implanted into the oviducts of pseudopregnant female mice. 28 pups were born, and 6 of them had at least one allele with desired loxp insertion with two of them being homozygous for recombinant alleles. We picked one founder to expand the colony. Genotyping was performed with a forward primer 5'- CTGCAACCACACACCAAGAA-3' and a reverse primer 5'- GCGT GACCTTATTAAGCTGCA-3', which provide 196 bp bands for wild type alleles and 236 bp bands for flox alleles.

## Histology

For morphometry studies, lungs were inflated with ice-cold 4% paraformaldehyde (PFA) under constant pressure of 25 cm H$_2$O for 5 minutes. Lungs were carefully dissected of attached structures and transferred into vial containing cold 4% PFA. Samples were rocked in 4 °C overnight, washed 3 x with PBS, and dehydrated in a series of ethanol (30%, 50% and 70%). Tissues were submitted in 70% ethanol to the UCSF Gladstone Histology and Light Microscopy core for further processing for paraffin embedding and tissue blocks were sectioned for Hematoxylin and Eosin (H&E) staining. For quantification of mean linear intercept (MLI), 6–8 sections of 5 μm thickness were sampled at 20 μm

levels through each set of lungs. Images were acquired on a Nikon Ti Inverted Microscope using a 10 x objective and DS-Ri2 color camera. For each sample, approximately 30 images were acquired from which 12 were randomly selected for MLI quantification using ImageJ software as described previously (*Crowley et al., 2019*). All samples were blinded during imaging and MLI quantification.

## Lung physiology

Pulmonary compliance and elastance were analyzed using the flexiVent system (SCIREQ) as previously described (*Tsujino et al., 2017*). Mice were anesthetized with ketamine (100 mg/kg), xylazine (10 mg/kg), and acepromazine (3 mg/kg) before a tracheostomy was performed to cannulate the trachea with a 20-guage catheter. Mice were then paralyzed with pancuronium (0.1 mg/kg) and analyzed using the flexiVent rodent ventilator. All mice were analyzed in a blinded fashion.

## Tissue dissociation

Mouse lungs were harvested after perfusion through the right ventricle with PBS. Lungs were dissected into individual lobes, minced with razor blades, and suspended in protease solution 0.25% Collagenase A (Millipore Sigma), 1 U/ml Dispase II (Millipore Sigma), 2000 U/ml Dnase I (Millipore Sigma) in Dulbecco's Modified Eagle Medium (Thermo Fisher) containing 10 mM HEPES (Millipore Sigma) and 2% FBS (Millipore Sigma). The suspension was incubated in a 37 °C water bath in a 15 ml conical for 25 min with aggressive trituration by glass Pasteur pipette every 8 minutes. The digestion was then quenched for 5 minutes in ice-cold PBS containing 5% FBS and 2 mM EDTA before passing cells through a 70 μm cell strainer. Cells were pelleted and resuspended in Red Blood Cell Lysing Buffer HybriMax (Sigma) for 5 minutes. RBC lysis was quenched dropwise with 5% FBS/PBS and cells were passed through a 40 μm cell strainer, washed, and resuspended in 5% FBS/PBS containing antibodies for FACS purification.

## Flow cytometry and fluorescence-activated cell sorting (FACS)

After tissue dissociation, cells were resuspended in 5% FBS/PBS with Fc-blocking antibody (TruStain FcX; BioLegend) at 0.01 mg/ml for 5 minutes at room temperature. All antibodies were resuspended in Brilliant Stain Buffer (BD Biosciences) and added to Fc-blocked cells for final antibody concentration of 1:200. Cells were stained for 30 min on ice for surface antibody staining, washed, and then passed through another 40 μm before analysis and/or cell sorting. DRAQ7 (BioLegend) was used 1:1000 to identify dead cells. Flow cytometric cell counting was performed using CountBright Plus Absolute Counting Beads (Thermo Fisher). The following antibodies were used in this study: anti-CD9 (clone MZ3, PE; BioLegend), anti-CD31 (clone 390, A488, BV605; BioLegend, BD Biosciences), anti-CD45 (clone 30 F-11, BV786, APC/Cy7, PE/Cy7; BioLegend, BD Biosciences), anti-Epcam (clone G8.8, FITC, PE, PE/Cy7; BioLegend), anti-I-A/I-E (clone M5/114.15.2, APC/Cy7, Spark UV 387; BioLegend), anti-Mcam (clone ME-9F1, A488, BUV496; BioLegend, BD Biosciences), anti-Pdgfra (clone APA5, APC, BV421, PE/Cy7; BioLegend). For proliferation studies, cells were stained for surface markers as described above and then washed, fixed, permeabilized, and processed for Click-iT EdU detection according to manufacturer's protocol (Click-iT Plus EdU Alexa Fluor 647 Flow Cytometry Assay Kit; Thermo Fisher). All samples were analyzed and sorted using an Aria Fusion (Becton Dickinson) with 85 μm nozzle except for scRNA-seq samples, which were collected using a FACS Aria III (Becton Dickinson) with 100 μm nozzle. Flow cytometry data were analyzed using FlowJo v10.8 (Becton Dickinson).

## Single-cell RNA-seq library preparation and sequencing

Lung tissues were harvested from *Tgfbr2$^{F/F}$* and *Tgfbr2$^{F/F}$*;Nkx2.1-cre mice at P7 and P14 after hyperoxia or normoxia treatment. Two pups were harvested for each genotype, exposure, and timepoint for total of 8 mice on P7 and 8 mice on P14. Single-cell suspension was obtained as described above. To enrich for epithelial and mesenchymal populations for sequencing input, 1x10$^5$ CD45+ cells, 1x10$^5$ CD31+ cells, and 1x10$^6$ CD45-/CD31- cells were sorted for each sample and collected in 10% FBS/PBS. Sorted cells were then counted and labeled with oligonucleotide tags for multiplexing using 10 x Genomics 3' CellPlex Kit Set A. All 8 biologic samples for the P7 timepoint were pooled and 60,000 cells / lane were loaded onto 2 lanes of a Chromium Next GEM Chip (10 x Genomics). The same workflow was used for P14 samples with all 8 biologic samples for P14 pooled together and loaded onto 2 lanes as well. Lanes 1 and 2 were therefore technical replicates for each biologic sample

and similarly lanes 3 and 4 were also technical replicates. Chromium Single Cell 3' v3.1 (10x Genomics) reagents were used for library preparation according to the manufacturer's protocol. The libraries were sequenced on an Illumina NovaSeq 6000 S4 flow cell.

## Sequencing data processing

Fastq files were uploaded to the 10 x Genomics Cloud (https://www.10xgenomics.com/products/cloud-analysis) and reads were aligned to the mouse reference genome mm10. The data were demultiplexed, and cells with multiple oligonucleotide tags were identified as multiplets and removed by the 10 x Genomics cloud analysis function with default parameters. Raw count matrices were imported to the R package Seurat v4.1.1 *Hao et al., 2021*, and cells with fewer than 200 detected genes, more than 7500 detected genes, or more than 15% mitochondria genes were excluded. We used the DoubletFinder package *McGinnis et al., 2019* for each sample using an estimated multiple rate of 1% remove doublets that were not detected upon alignment. We then merged all the sample objects, identified the top variable genes using Seurat's *FindVariableGenes* function, and integrated the samples using the *RunFastMNN* function of the SeuratWrappers R package (*Butler et al., 2018*). For visualization, Seurat's *RunUMAP* function was performed using MNN dimensional reduction.

Twenty-three clusters were initially identified from a total of 27,839 cells using Seurat's *FindNeighbors* and *FindClusters* functions with resolution = 0.3. The expression of canonical linage markers (Epcam, Col1a1, Pecam1, Ptprc, Msln) was used to define major cell types of the lung (epithelium, mesenchyme, hematopoietic, endothelium, mesothelium). Cluster 9 (729 cells) was identified as a contaminant and excluded from further analysis due to cells showing up in multiple locations across the UMAP embedding. While re-clustering and annotating epithelial, hematopoietic, and endothelial subpopulations we identified an additional 500 cells that clustered independently from known cell types within each subcluster, had less than 1000 detected genes, and were enriched for the expression of multiple canonical cell types, suggesting these cells were also contaminants. After removing these cells and cluster 9 cells, the remaining 26,610 cells were re-clustered with *FindVariableGenes*, *RunFastMNN*, *RunUMAP*, *FindNeighbors*, and *FindClusters* functions with clustering resolution = 0.8. Thirty clusters were identified at this stage and differentially expressed genes for each cluster were identified using *FindAllMarkers* focusing on genes expressed by more than 20% of cells (either within or outside of a cluster) and with a log fold change greater than 0.2. Using publicly available data, we were able to merge and annotate clusters to obtain the 25 clusters depicted in this study. Mesenchymal cells were re-clustered using the same workflow outlined above with clustering resolution = 0.3. Clusters were annotated based on a combination of three publicly available data sets for the neonatal lung.

For Qiagen Ingenuity Pathway Analysis (QIAGEN, https://digitalinsights.qiagen.com/IPA) (*Krämer et al., 2014*), differentially expressed genes in myofibroblasts were identified by comparing either CTRL RA vs O2 cells or RA CTRL vs cKO cells using Seurat's *FindMarkers* function. These lists were uploaded to Qiagen IPA to identify predicted upstream regulators for each comparison. To identify shared predicted upstream regulators, lists were filtered for z-score>1.5 (upregulated) or z-score<−1.5 (downregulated), and the filtered results were compared by Venn diagram between the two comparisons.

NicheNet (*Browaeys et al., 2020*) was used to compare signaling between conditions, as described in the 'Differential NicheNet analysis between conditions of interest' vignette on the GitHub repository (*Browaeys and Sang-Aram, 2024*). Standard NicheNet statistical thresholds described in the vignette were used in the analysis. Alveolar epithelial subtypes (AT2, AT2 Lyz1, AT2 Activated, AT1/AT2, and Prolif. AT2) were used as a combined sender cell population, and alveolar and ductal myofibroblasts were used as combined receiver cell populations. Differential signaling from sender-to-receiver cells was compared across two conditions, CTRL RA vs CTRL O2 and CTRL RA vs cKO RA, and the results were examined for pathways present in both comparisons.

## Data analysis

Mean linear intercept morphometry was quantified using ImageJ software as described previously (*Crowley et al., 2019*). scRNA-seq data analysis was performed in R version 4.1.1. Statistical tests were performed in GraphPad Prism version 9.4.0.

## Acknowledgements

We thank members of the Sheppard lab for helpful advice and discussion. We thank the following UCSF core facilities for their technical support: Laboratory for Cell Analysis (supported by P30CA082103), Center for Advanced Light Microscopy (supported by UCSF PBBR), Gladstone Genomics Core, Gladstone Histology and Light Microscopy Core, Gladstone Transgenic Gene Targeting Core, and the Center for Advanced Technology (supported by UCSF PBBR, RRP IMIA, and 1S10OD028511-01). ISK was supported by the Pediatric Scientist Development Program, the Nina Ireland Program for Lung Health, and the UCSF Division of Neonatology. This work was supported by HD000850 (NICDH, ISK); UCSF Division of Neonatology (ISK); and HL145037 and HL142568 (NHLBI, DS).

## Additional information

### Funding

| Funder | Grant reference number | Author |
|---|---|---|
| National Institute of Child Health and Human Development | Pediatric Scientist Development Program | Imran S Khan |
| National Institute of Child Health and Human Development | HD000850 | Imran S Khan |
| University of California, San Francisco | Nina Ireland Program For Lung Health | Imran S Khan |
| University of California, San Francisco | | Imran S Khan |
| National Heart, Lung, and Blood Institute | HL145037 | Dean Sheppard |
| National Heart, Lung, and Blood Institute | HL142568 | Dean Sheppard |

The funders had no role in study design, data collection and interpretation, or the decision to submit the work for publication.

### Author contributions

Imran S Khan, Conceptualization, Data curation, Formal analysis, Funding acquisition, Investigation, Methodology, Writing – original draft, Writing – review and editing; Christopher Molina, Investigation, Writing – review and editing; Xin Ren, Investigation; Vincent C Auyeung, Data curation, Formal analysis, Investigation, Writing – review and editing; Max Cohen, Data curation, Formal analysis; Tatsuya Tsukui, Resources, Formal analysis, Investigation; Amha Atakilit, Resources, Methodology; Dean Sheppard, Conceptualization, Resources, Supervision, Funding acquisition, Investigation, Methodology, Project administration, Writing – review and editing

### Author ORCIDs

Imran S Khan ⓘ https://orcid.org/0000-0003-4570-4143
Vincent C Auyeung ⓘ https://orcid.org/0000-0001-6273-1595
Tatsuya Tsukui ⓘ https://orcid.org/0000-0003-3100-6934
Dean Sheppard ⓘ https://orcid.org/0000-0002-6277-2036

### Ethics

This study was performed in strict accordance with the recommendations in the Guide for the Care and Use of Laboratory Animals of the National Institutes of Health. All of the animals were handled according to approved institutional animal care and use committee (IACUC) of the University of California San Francisco. The protocol was approved by IACUC (protocol AN195465). All surgery was performed under isoflurane anesthesia, and every effort was made to minimize suffering.

Reviewer #1 (Public review): https://doi.org/10.7554/eLife.94425.3.sa1
Reviewer #2 (Public review): https://doi.org/10.7554/eLife.94425.3.sa2
Reviewer #3 (Public review): https://doi.org/10.7554/eLife.94425.3.sa3
Author response https://doi.org/10.7554/eLife.94425.3.sa4

## Additional files

### Supplementary files
• MDAR checklist

### Data availability

Sequencing data and Seurat objects have been deposited in Gene Expression Omnibus Series GSE243129.

The following dataset was generated:

| Author(s) | Year | Dataset title | Dataset URL | Database and Identifier |
|---|---|---|---|---|
| Khan I, Sheppard D | 2024 | Single-cell RNA Sequencing of Murine Lungs to Compare Two Different Models of Alveolar Simplification | https://www.ncbi.nlm.nih.gov/geo/query/acc.cgi?acc=GSE243129 | NCBI Gene Expression Omnibus, GSE243129 |

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
