## [Editor Report · eLife Assessment]

This study provides **important** insights into postnatal lung development and the mechanisms underlying bronchopulmonary dysplasia (BPD), a condition with high morbidity and mortality in newborns. Through the use of neonatal hyperoxia, cell-type-specific inactivation of Tgfbr2, and other injury models, the research focuses on the role of TGF-β signaling in BPD pathogenesis, highlighting impaired myofibroblast proliferation as a key factor. The inactivation of Etc2 in Pdgfra-lineaged cells disrupts myofibroblast cytokinesis, leading to alveolar simplification and reduced cell numbers. The use of transgenic mice and single-cell transcriptomics offers a detailed and high-quality dataset, advancing our understanding of BPD and serving as a invaluable resource for developmental biology and neonatal pulmonary research. The study's comprehensive approach, robust data, and methodological rigor make it a **compelling** contribution to the field, providing both mechanistic insights and a resource for further research into BPD pathogenesis.

---

## [Referee Report · Reviewer #1 (Public review)]

Summary:

In this study, the authors used both the commonly used neonatal hyperoxia model as well as cell-type-specific genetic inactivation of Tgfbr2 models to study the basis of BPD. The bulk of the analyses focus on the mesenchymal cells. Results indicate impaired myofibroblast proliferation, resulting in decreased cell number. Inactivation of Etc2 in Pdgfra-lineaged cells, preventing cytokinesis of myofibroblasts, led to alveolar simplification. Together, the findings demonstrate that disrupted myofibroblast proliferation is a key contributor to BPD pathogenesis.

Strengths:

Overall, this comprehensive study of BPD models advances our understanding of the disease. The data are of high quality.

Comments on latest version:

In the revision, the authors addressed all critiques.

---

## [Referee Report · Reviewer #2 (Public review)]

Summary:

In this study the authors systematically explore mechanism(s) of impaired postnatal lung development with relevance to BPD (bronchopulmonary dysplasia) in two murine models of 'alveolar simplification', namely hyperoxia and epithelial loss of TGFb signaling. The work presented here is of great importance, given the limited treatment options for a clinical entity frequently encountered in newborns with high morbidity and mortality that is still poorly understood, and the unclear role of TGFb signaling, its signaling levels, and its cellular effects during secondary alveolar septum formation, a lung structure generating event heavily impacted by BPD. The authors show that hyperoxia and epithelial TGFb signaling loss have similar detrimental effects on lung structure and mechanical properties (emphysema-like phenotype) and are associated with significantly decreases numbers of PDGFRa-expressing cells, the major cell pool responsible for generation of postnatal myofibroblasts. They then use a single-cell transcriptomic approach combined with pathway enrichment analysis for both models to elucidate common factors that affect alveologenesis. Using cell communication analysis (NicheNet) between epithelial and myofibroblasts they confirm increased projected TGFb-TGFbR interactions and decreased projected interactions for PDGFA-PDGFRA, and other key pathways, such as SHH and WNT. Based on these results they go on to uncover in a sequela of experiments that surprisingly, increased TGFb appears reactive to postnatal lung injury and rather protective/homeostatic in nature, and the authors establish the requirement for alpha V integrins, but not the subtype alphaVbeta6, a known activator of TGFb signaling and implied in adult lung fibrosis. The authors then go beyond the TGFb axis evaluation to show that mere inhibition of proliferation by conditional KO of Ect2 in Pdgfra lineage results in alveolar simplification, pointing out the pivotal role of PDGFRa-expressing myofibroblasts for normal postnatal lung development.

Strengths:

(1) The approach including both pharmacologic and mechanistically-relevant transgenic interventions both of which produced consistent results provides robustness of the results presented here.

(2) Further adding to this robustness is the use of moderate levels of hyperoxia at 75% FiO2, which is less extreme than 100% FiO2 frequently used by others in the field, and therefore favors the null hypothesis.

(3) The prudent use of advancement single cell analysis tools, such as NicheNet to establish cell interactions through the pathways they tested and the validation of their scRNA-seq results by analysis of two external datasets. Delineation of the complexity of signals between different cell types during normal and perturbed lung development, such as attempted successfully in this study, will yield further insights into the underlying mechanism(s).

(4) The combined readout of lung morphometric (MLI) and lung physiologic parameters generates a clinically meaningful readout of lung structure and function.

(5) The systematic evaluation of TGFb signaling better determines the role in normal and postnatally-injured lung.

Weaknesses:

(1) While the study convincingly establishes the effect of lung injury on the proliferation of PDGFRa-expressing cells, differentiation is equally important. Characterization of PDGFRa expressing cells and tracking the changes in the injury models in the scRNA analysis, a key feature of this study, would benefit from expansion in this regard. PDGFRa lineage gives rise to several key fibroblast populations, including myofibroblasts, lipofibroblasts, and matrix-type fibroblasts (Collagen13a1, Collagen14a1). Lipofibroblasts constitute a significant fraction of PDGFRa+ cells, and expand in response to hyperoxic injury, as shown by others. Collagen13a1-expressing fibroblasts expand significantly under both conditions (Fig.3), and appear to contain a significant number of PDGFRa-expressing cells (Suppl Fig.1). Effects of the applied injuries on known differentiation markers for these populations should be documented. Another important aspect would be to evaluate whether the protective/homeostatic effect of TGFb signaling is by supporting differentiation of myofibroblasts. Postnatal Gli1 lineage gains expression of PDGFRa and differentiation markers, such as Acta2 (SMA) and Eln (Tropoelastin). Loss of PDGFRa expression was shown to alter Elastin and TGFb pathway related genes. TGFb signaling is tightly linked to the ECM via LTBPs, Fibrillins and Fibulins. An additional analysis in the aforementioned regards has great potential to more specifically identify the cell type(s) affected by the loss of TGFb signaling and allow analysis of their specific transcriptomic changes in response and underlying mechanism(s) to postnatal injury.

[The authors have added in detailed transcriptomic description of the fibroblast populations.]

(2) Of the three major lung abnormalities encountered in BPD, the authors focus on alveolarization impairment in great detail, to very limited extend on inflammation, and not on vascularization impairment. However, this would be important not only to better capture the established pathohistologic abnormalities of BPD, but also is needed since the authors alter TGFb signaling, and inflammatory and vascular phenotypes with developmental loss of TGFb signaling and its activators have been described. Since the authors make the point about absence of inflammation in their BPD model, it will be important to show the evidence.

[While this an important question, assessment of these components goes beyond the scope of this paper.]

(3) Conceptually it would be important that in the discussion the authors reconcile their findings in the experimental BPD models in light of human BPD and potential implications it might have on new ways to target key pathways and cell types for treatment. This allows the scientific community to formulate the next set of questions in a disease relevant manner.

[The authors have amended the discussion in this regard.]

Comments on latest version:

This reviewer would like to thank the authors for their efforts to address the concerns, in particular the better transcriptomic description of the fibroblast populations. The reviewer is well aware of the issues with PDGFRa antibodies that work on mouse tissue and also the problem with available reporters and lineage tracers in terms of haploinsufficiency.

There are no further concerns from this reviewer's side.

---

## [Referee Report · Reviewer #3 (Public review)]

This paper seeks to understand the role of alveolar myofibroblasts in the abnormal lung development after saccular stage injury.

Strengths:

(1) Multiple models of neonatal injury are used, hyperoxia and transgenic models that target alveolar myofibroblasts.

(2) The authors integrate their data with prior published single-cell data from neonatal hyperoxia injury models and demonstrate concordant findings.

Weaknesses:

(1) As the authors acknowledge in the discussion, there are no spatial and temporal validation data of the single-cell findings. As the ductal myofibroblasts has many overlapping genes, localizing and quantifying the loss of these cells in injury as a plausible mechanistic driver would greatly strengthen the conclusion.

(2) As they note in their response, this proved to be technically difficult and current Pdgfra-lineage trace tools are not without their own limitations.

Summary:

Taken together, this manuscript provides a rich data set from a model of irreversible neonatal lung injury. The single-cell analysis methods are well-articulated and the limitations are acknowledged, allowing this paper to provide a foundation for future work to spatially and temporally validate these claims.

---

## [Author Response]

The following is the authors’ response to the original reviews.

We have responded to these criticisms below and have revised the main text and figures. Here, we outline the major points of our responses:

(1) The reviewers asked for more clarification regarding cell type annotation in the lung mesenchyme as shown in Figure 3C. We have included a new supplementary figure (Supplementary Figure 2) which shows differentially expressed genes amongst these mesenchymal cell subsets using a variety of visualization tools including a heatmap, UMAP plots, and the dotplot which was originally shown in Supplementary Figure 1D. The other supplemental figures have been re-numbered.

(2) We acknowledge the lack of consensus in the field regarding the nomenclature of fibroblast subsets in the developing mouse lung. We are not attempting to define new subsets, but rather we adopted annotations based on previously published work. Specifically, we used Seurat to define mesenchymal cell clusters and then compared the gene expression patterns of these clusters to published work by Hurskainen et al. (Bernard Thebaud’s group) and Narvaez Del Pilar et al. (Jichou Chen’s group). We acknowledge these annotations might conflict with other published data, but any approach to choosing a cell label would be subject to scrutiny. For example, Col13a1 fibroblasts share markers with cells which have been defined by others as lipofibroblasts or alveolar fibroblasts. Similarly, Col14a1 fibroblasts appear to share markers with matrix fibroblasts. Further work is clearly needed to address these discrepancies, and we hope that making our data publicly available will help that effort.

(3) The reviewers asked us to interrogate changes in canonical markers of fibroblast subsets (i.e. lipofibroblasts, matrix fibroblasts) to address whether the apparent loss of myofibroblasts could be explained by a change in myofibroblast specification/differentiation. We have included these data in the responses, but because we are unable to draw any clear conclusions from these results, we do not feel these data warrant inclusion in the manuscript/figures.

(4) As highlighted in the eLife assessment, our study does not include tissue validation (i.e. immunohistochemistry) of myofibroblast markers to distinguish whether the loss of myofibroblasts is attributable to lack of proliferation and/or changes in differentiation/specification. We spent considerable time over the past few months attempting to address these questions, however we were unable to produce convincing PDGFRa staining on tissues that we had collected during our original studies. Without PDGFRa staining, we regretfully could not co-stain for other useful markers to assess proliferation (EdU), apoptosis (TUNEL or caspase), or fibroblast function/specification (ACTA2, SM22a/TAGLN, ADRP, etc). We suspect that these experiments would require optimization of tissue fixation/processing at the time of harvest or the inclusion of a *Pdgfra* lineage tool for better identification of these cells by immunohistochemistry. Given that the majority of *Pdgfra* lineage tools require a knock-in/knock-out approach, data generated using these tools should be interpreted with caution given our results here show that *Pdgfra*-haploinsufficiency alone worsens disease outcomes after hyperoxia exposure.

In summary, we have addressed several concerns raised by the reviewers and have attempted to perform some of the additional experiments suggested.

**Public Reviews:**

**Reviewer #1 (Public Review):**
Summary:In this study, the authors used both the commonly used neonatal hyperoxia model as well as cell-type-specific genetic inactivation of Tgfbr2 models to study the basis of BPD. The bulk of the analyses focus on the mesenchymal cells. Results indicate impaired myofibroblast proliferation, resulting in decreased cell number. Inactivation of Etc2 in Pdgfra-lineaged cells, preventing cytokinesis of myofibroblasts, led to alveolar simplification. Together, the findings demonstrate that disrupted myofibroblast proliferation is a key contributor to BPD pathogenesis.Strengths:Overall, this comprehensive study of BPD models advances our understanding of the disease. The data are of high quality.Weaknesses:The critiques are mostly minor and can be addressed without extensive experimentation.
**Reviewer #2 (Public Review):**
Summary:In this study, the authors systematically explore the mechanism(s) of impaired postnatal lung development with relevance to BPD (bronchopulmonary dysplasia) in two murine models of 'alveolar simplification', namely hyperoxia and epithelial loss of TGFb signaling. The work presented here is of great importance, given the limited treatment options for a clinical entity frequently encountered in newborns with high morbidity and mortality that is still poorly understood, and the unclear role of TGFb signaling, its signaling levels, and its cellular effects during secondary alveolar septum formation, a lung structure generating event heavily impacted by BPD. The authors show that hyperoxia and epithelial TGFb signaling loss have similar detrimental effects on lung structure and mechanical properties (emphysema-like phenotype) and are associated with significantly decreased numbers of PDGFRa-expressing cells, the major cell pool responsible for generation of postnatal myofibroblasts. They then use a single-cell transcriptomic approach combined with pathway enrichment analysis for both models to elucidate common factors that affect alveologenesis. Using cell communication analysis (NicheNet) between epithelial and myofibroblasts they confirm increased projected TGFb-TGFbR interactions and decreased projected interactions for PDGFA-PDGFRA, and other key pathways, such as SHH and WNT. Based on these results they go on to uncover in a sequela of experiments that surprisingly, increased TGFb appears reactive to postnatal lung injury and rather protective/homeostatic in nature, and the authors establish the requirement for alpha V integrins, but not the subtype alphaVbeta6, a known activator of TGFb signaling and implied in adult lung fibrosis. The authors then go beyond the TGFb axis evaluation to show that mere inhibition of proliferation by conditional KO of Ect2 in Pdgfra lineage results in alveolar simplification, pointing out the pivotal role of PDGFRa-expressing myofibroblasts for normal postnatal lung development.Strengths:(1) The approach including both pharmacologic and mechanistically-relevant transgenic interventions both of which produced consistent results provides robustness of the results presented here.(2) Further adding to this robustness is the use of moderate levels of hyperoxia at 75% FiO2, which is less extreme than 100% FiO2 frequently used by others in the field, and therefore favors the null hypothesis.(3) The prudent use of advanced single-cell analysis tools, such as NicheNet to establish cell interactions through the pathways they tested and the validation of their scRNA-seq results by analysis of two external datasets. Delineation of the complexity of signals between different cell types during normal and perturbed lung development, such as attempted successfully in this study, will yield further insights into the underlying mechanism(s).(4) The combined readout of lung morphometric (MLI) and lung physiologic parameters generates a clinically meaningful readout of lung structure and function.(5) The systematic evaluation of TGFb signaling better determines the role in normal and postnatally-injured lungs.Weaknesses:(1) While the study convincingly establishes the effect of lung injury on the proliferation of PDGFRa-expressing cells, differentiation is equally important. Characterization of PDGFRa expressing cells and tracking the changes in the injury models in the scRNA analysis, a key feature of this study, would benefit from expansion in this regard. PDGFRa lineage gives rise to several key fibroblast populations, including myofibroblasts, lipofibroblasts, and matrix-type fibroblasts (Collagen13a1, Collagen14a1). Lipofibroblasts constitute a significant fraction of PDGFRa+ cells, and expand in response to hyperoxic injury, as shown by others. Collagen13a1-expressing fibroblasts expand significantly under both conditions (Figure 3), and appear to contain a significant number of PDGFRa-expressing cells (Suppl Fig.1). Effects of the applied injuries on known differentiation markers for these populations should be documented. Another important aspect would be to evaluate whether the protective/homeostatic effect of TGFb signaling is supporting the differentiation of myofibroblasts. Postnatal Gli1 lineage gains expression of PDGFRa and differentiation markers, such as Acta2 (SMA) and Eln (Tropoelastin). Loss of PDGFRa expression was shown to alter Elastin and TGFb pathway-related genes. TGFb signaling is tightly linked to the ECM via LTBPs, Fibrillins, and Fibulins. An additional analysis in the aforementioned regard has great potential to more specifically identify the cell type(s) affected by the loss of TGFb signaling and allow analysis of their specific transcriptomic changes in response and underlying mechanism(s) to postnatal injury.

We attempted to conduct additional analyses on our sequencing data to evaluate the impact of lung injury on the differentiation of *Pdgfra*-expressing cells towards other fibroblast lineages. To specifically address the impact of hyperoxia on fibroblast differentiation, we subsetted wildtype cells collected at the P7 timepoint (while pups were still undergoing hyperoxia treatment) from the larger data set. Shown below are several Violin Plots comparing gene expression between RA and O2 conditions across the mesenchymal populations.

Although there are some interesting observations in this analysis, we could not identify a consistent theme from these data which could clearly answer the reviewers’ questions. We see a clear reduction of *Pdgfra* and *Eln* in both myofibroblast subsets with hyperoxia, which support our findings of reductions in the myofibroblast subsets. *Acta2* and *Tagln* appear slightly lower in alveolar myofibroblasts, but both are higher in ductal myofibroblasts. Interestingly, both *Acta2* and *Tagln* are higher in Col14a1 fibroblasts with hyperoxia. The functional relevance of these data are unclear because there appears to be higher per-cell expression of *Acta2* in ductal myofibroblasts while the relative contribution of these cells is reduced (Figure 3D-E). Col14a1 fibroblasts show increased *Acta2* and *Tagln* expression and are slightly increased in proportion at P7 with hyperoxia treatment (Figure 3D), albeit to a much lesser degree compared to Col13a1 fibroblasts.

**Author response image 1. sa4fig1:** 

Markers of ductal myofibroblasts including *Hhip*, *Cdh4*, and *Aspn* all appear lower with hyperoxia. Interestingly *Plin2* expression is only slightly increased in Col13a1 fibroblasts with hyperoxia treatment, and there is also increased expression in alveolar myofibroblasts. *Tcf21* is another marker commonly used to identify lipofibroblasts and its expression is similarly increased in myofibroblasts during hyperoxia, although its expression is conversely lower in Col13a1 and Col14a1 fibroblasts in our data. Overall, these data would appear consistent with recently published data by Ricetti *et al*. in which the authors observed an increase in lipofibroblast gene signatures and reduced myofibroblast gene signatures with hyperoxia treatment.

**Author response image 2. sa4fig2:** 

**Author response image 3. sa4fig3:** 

The ability of our data to clearly identify changes in cell fate differentiation is limited by our use of Seurat to define cell clusters because these methods are likely to mask subtle gene expression changes in a small number of cells nested within a parent cluster. In the example above with *Plin2*, the change in Plin2 expression within myofibroblasts is not significant enough for Seurat to pull these cells out from their parent clusters to define a different lineage, nor are these cells similar enough in their current moment in time to be considered Col13a1 fibroblasts or lipofibroblasts. Increasing the dimensions used to define Seurat clusters might be sufficient to identify this subset of cells as a distinct cluster, however this approach would come at the expense of creating several more cell subsets with increasingly small populations which would be difficult to further analyze.

One alternative approach to address these questions regarding differentiation might include using pseudo-time analysis of our sequencing data to predict cell lineage. Unfortunately, these analyses are beyond the scope of our current study, but we hope that our public data set can be used by investigators hoping to utilize this approach. Another method to address these questions could utilize a pulse-chase lineage experiment where one could label *Pdgfra*-expressing cells at the onset of injury and compare the differentiation of these labeled cells following injury. Li *et al*. conducted a similar experiment with hyperoxia in which *Pdgfra*-expressing cells were labeled during embryonic development and then postnatally following hyperoxia exposure. The authors noted a decrease in both lineaged myofibroblasts and lineaged lipofibroblasts and concluded that *Pdgfra*-lineaged cells were lost with hyperoxia treatment rather than undergoing aberrant differentiation. While these experiments likely have their own caveats related to the timing and efficiency of labeling, they represent a more conclusive approach to addressing differences in cell specification as compared to our sequencing- and flow cytometry-based approaches.

**Author response image 4. sa4fig4:** 

**Author response image 5. sa4fig5:** 

(2) Of the three major lung abnormalities encountered in BPD, the authors focus on alveolarization impairment in great detail, to a very limited extent on inflammation, and not on vascularization impairment. However, this would be important not only to better capture the established pathohistologic abnormalities of BPD, but also it is needed since the authors alter TGFb signaling, and inflammatory and vascular phenotypes with developmental loss of TGFb signaling and its activators have been described. Since the authors make the point about the absence of inflammation in their BPD model, it will be important to show the evidence.

We acknowledge that vascular changes significantly contribute to BPD pathogenesis, however our study was not designed to adequately characterize changes in vascular/endothelial cells. We were motivated to focus on the lung mesenchyme after observing a dramatic loss of PDGFRa+ cells with our initial characterization of the hyperoxia injury model (Figure 2). At the onset of our study, the existing publicly available data did not contain enough mesenchymal cells for in-depth analysis. To generate new observations and hypotheses within the lung mesenchyme we enriched our single cell prep for mesenchymal cells at the time of FACS-sorting to ensure we would have sufficient cell numbers for downstream analysis.

(3) Conceptually it would be important that in the discussion the authors reconcile their findings in the experimental BPD models in light of human BPD and the potential implications it might have on new ways to target key pathways and cell types for treatment. This allows the scientific community to formulate the next set of questions in a disease-relevant manner.

We have edited text in the discussion to address this point.

**Reviewer #3 (Public Review):**
Summary:This paper seeks to understand the role of alveolar myofibroblasts in abnormal lung development after saccular stage injury.Strengths:Multiple models of neonatal injury are used, including hyperoxia and transgenic models that target alveolar myofibroblasts.Weaknesses:There are several weaknesses that leave the conclusions significantly undersupported by the data as presented:(1) There is no validation of the decreased number of myofibroblasts suggested by flow cytometry/scRNAseq at the level of the tissue. Given that multiple groups have reported increased myofibroblasts (aSMA+ fibroblasts) in humans with BPD and in mouse models, demonstrating a departure from prior findings with tissue validation in the mouse models is essential. There are many reasons for decreased numbers of a subpopulation by flow cytometry, most notably that injured cells may be less likely to survive the cell sorting process.

Unfortunately, we were unable to produce convincing PDGFRa staining on tissues that we had collected during our original studies. Without PDGFRa staining, we regretfully could not co-stain for other useful markers to assess proliferation (EdU), apoptosis (TUNEL or caspase), or fibroblast function/specification (aSMA/ACTA2, SM22a/TAGLN, ADRP, etc). We suspect that these experiments would require optimization of tissue fixation/processing at the time of harvest or the inclusion of a *Pdgfra* lineage tool for better identification of these cells by immunohistochemistry. Given that the majority of *Pdgfra* lineage tools require a knock-in/knock-out approach, data generated using these tools should be interpreted with caution given our results here show that *Pdgfra*-haploinsufficiency alone worsens disease outcomes after hyperoxia exposure.

Our single cell data show that there is increased expression of *Acta2* and *Tagln* shown in the plots which might be consistent with the increased aSMA staining which others have observed in these settings. Interestingly, the transcripts of both genes are reduced in alveolar fibroblasts while increased in ductal myofibroblasts, Col13a1 fibroblasts, Col14a1 fibroblasts, and vascular smooth muscle. We did not include aSMA antibody staining in our flow cytometry experiments, but this would certainly add value to future attempts to characterize the phenotypic changes occurring during these injury models.

(2) The hallmark genes used to define the subpopulations are not given in single-cell data. As the definition of fibroblast subtypes remains an area of unsettled discussion in the field, it is possible that the decreased number by classification and not a true difference. Tissue validation and more transparency in the methods used for single-cell sequencing would be critical here.

See response above and new Supplemental Figure 2.

(3) There is an oversimplification of neonatal hyperoxia as a "BPD model" used here without a reference to detailed prior work demonstrating that the degree and duration of hyperoxia dramatically change the phenotype. For example, Morty et al have shown that hyperoxia of 85% or more x 14 days is required to demonstrate the septal thickening observed in severe human BPD. Other than one metric of lung morphometry (MLI), which is missing units on the y-axis and flexivent data, the authors have not fully characterized this model. Prior work comparing 75% O2 exposure for 5, 8, or 14 days shows that in the 8-day exposed group (similar to the model used here), much of the injury was reversible. What evidence do the authors have that hyperoxia alone is an accurate model of the permanent structural injury seen in human BPD?

At the onset of our studies, we noted that several groups were using widely variable protocols ranging from 60-100% O2 exposure. Morty *et al.* have indeed conducted thorough experiments to characterize various different hyperoxia exposure protocols. In their 2017 study (https://www.ncbi.nlm.nih.gov/pmc/articles/PMC5312005/) they showed that 85% O2 from P1-P7 was sufficient to produce increased septal thickness compared to control mice, and this change was comparable to P1-P14 exposure with 85% O2. Interestingly, they also noted that some therapeutic interventions could rescue disease caused by 60% O2 but not 85% O2 exposure. Our criteria in choosing a treatment protocol were: (1) nursing dams and pups survived hyperoxia exposure, (2) injury was reproducible across cohorts, and (3) injury was not reversible simply by recovering in room air. We found that recent work utilizing 75% O2 exposure was sufficient to cause the alveolar simplification phenotype which we sought to investigate. In our hands, we did not observe mortality of nursing dams or pups except for litters lost to cannibalism/failure of cross-fostering.

We are confident that the injury caused by our hyperoxia protocol is not reversible simply by recovering mice in room air. Several groups have phenotyped mice at P4, P10, or P14 immediately following the conclusion of hyperoxia treatment. To ensure that we were studying a lasting, irreversible phenotype, we conducted our endpoint studies (morphometry and lung physiology) at P40. Because mice continue to undergo alveolarization until ~P36-P39, we reasoned that this additional recovery time following cessation of hyperoxia would allow for spontaneous recovery if this injury was transient. Additionally, shown below are unpublished flexiVent data in which mice were treated for 10 days with 75% O2 and recovered until analysis at 10 weeks of age. These results are entirely consistent with the flexiVent data we have included in the manuscript, and the persistence of lung physiologic changes in adult mice suggest the presence of permanent underlying structural changes. We did not conduct morphometry/MLI studies at later timepoints, but we have no reason to suspect a different outcome given the clear results from lung physiology.

**Author response image 6. sa4fig6:** 

(4) Thibeault et al published a single-cell analysis of neonatal hyperoxia in 2021, with seemingly contrasting findings. How does this dataset compare in context?

Our data is complimentary to the single-cell analysis published by Thebaud *et al*. We included a re-analysis of their mesenchymal data in Supplementary Figure 2 which shows they also observed a relative decrease in myofibroblast clusters at the P7 and P14 timepoints following hyperoxia treatment. Figure 4 of their paper highlights the top differentially expressed genes between RA and O2 in Col13a1 FB and myofibroblasts, and we observe nearly identical findings in our data set within each of these clusters. Below we have created dotplots of P7 wildtype samples for the same selected genes shown in Figure 4G of the Thebaud *et al*. paper. It is important to note that their clustering pooled all myofibroblasts into one cluster, while our data is divided into alveolar myofibroblasts and ductal myofibroblasts. The other difference is their data set includes all timepoints P3, P7 and P14 pooled for display, while the plot we selected for simplicity here is only P7 cells. From these data we can see that the general trends are identical to those observed by Thebaud *et al*., and the differences in genes such as Acta2 can be accounted for by different changes observed in the different myofibroblast clusters – which is identical to what is shown in the violin plots above – namely that Acta2 is reduced in hyperoxia in alveolar myofibroblasts while increased in the ductal myofibroblasts.

**Author response image 7. sa4fig7:** Alveolar myoFB.

**Author response image 8. sa4fig8:** Ductal myoFB.

One difference between our two datasets is the relative contribution of myofibroblast and Col13a1 fibroblasts to the entire mesenchymal population of cells. Over 50% of all mesenchymal cells in our preps consist of myofibroblasts, while most of their mesenchymal cells are Col13a1 fibroblasts. These differences are likely accounted for by differences in tissue digestion and cell preparation protocols. However, despite these differences, their data show the same trends of decreased myofibroblasts and a relative expansion in Col13a1 fibroblasts.

**Recommendations for the authors:**

**Reviewer #1 (Recommendations For The Authors):**
(1) Figure 1, for the hyperoxia model, it is informative to have the analysis done at P40, while most of the previous studies using this model focus on outcomes shortly after the end of the hyperoxia regimen. The authors state "we did not see evidence of fibrosis, scarring, or inflammation." It will be helpful to include data supporting this conclusion, especially ACTA2, CTHRC1, and CD45 staining.

We did not conduct trichrome staining or hydroxyproline assays to quantify the absence of fibrotic changes because there were no gross histologic changes consistent with scarring or fibrosis by H&E staining. We have amended the text to say “we did not see evidence of fibrosis or scarring” since we did not publish any changes to characterize the immune cell compartment.

(2) Figure 3, single cell analysis, naming of the clusters is confusing. Is "alveolar myofibroblasts" the same as "secondary crest myofibroblasts"? Is "Col13a1 FB" the same as "alveolar fibroblasts" and "Col14a1 FB" the same as "adventitial fibroblasts"? The loss of myofibroblasts is intriguing because, by staining, there is an increase of ACTA2+ cells. Are ACTA2+ cells not myofibroblasts in scRNAseq data?

As mentioned in responses above, we used Jichou Chen’s nomenclature of “alveolar myofibroblasts” and “ductal myofibroblasts”, but we agree that the former cluster is most consistent with “secondary crest myofibroblasts”. To distinguish the two remaining clusters of fibroblasts we used the same nomenclature as found in Thebaud et al’s single cell data set- “Col13a1 FB and “Col14a1 FB”. The Col13a1 FB cluster is most consistent with “alveolar fibroblasts” and contains high expression of several genes used to define “lipofibroblasts”, though it is unclear whether the latter may represent a subcluster within the Col13a1 FB cluster.

As shown above, *Acta2* is expressed broadly within the lung mesenchyme with highest levels found in myofibroblasts and smooth muscle cells.

(3) Phosphorylated SMAD2/3 staining (e.g. Cell Signaling antibody) in the two models will be informative to show where TGF signaling activity is altered.

We have not been successful in using SMAD2/3 staining to infer changes in TGFb signaling at the resolution needed to address this question. Other groups have shown qPCR and western blot data for SMAD2/3 signaling from whole lung extracts, but these approaches lack cell type and specificity and do not address spatial changes. We attempted to incorporate pSMAD2/3 staining into our flow cytometry experiments, but the staining protocol did not work in our hands.

(4) Is cell death increased in the multiple models that showed simplification?

While our EdU experiments address proliferation, we were unable to perform PDGFRa and TUNEL/caspase co-staining by histology to address apoptosis/cell death in our different models. Shown here is data from P7 wildtype mice in which *Cdkn1a* (promoting arrest of cell cycle), and pro-apoptotic genes *Bax*, *Bak1*, and *Fas* are all upregulated in hyperoxia in several mesenchymal cell populations including myofibroblasts.

**Author response image 9. sa4fig9:** 

(5) Wording: "These data suggest that avb6 does not play a role in TGFb activation during normal development or neonatal hyperoxia, while av-integrins in the lung mesenchyme are required for normal development and play a protective role in response to hyperoxia." The first half of the sentence is missing a reference to the epithelium.

Text now reads "These data suggest that epithelial avb6 does not play a role…”

**Reviewer #2 (Recommendations For The Authors):**
The reviewer greatly appreciates the work presented here, especially the hard task of addressing combined signaling pathway input into key mesenchymal cell types during an essential expansion of alveolar surface area in postnatal lung and its effect upon disturbance.The issues of concern are mentioned in the public review and are expanded upon below:(1) Expanded characterization of PDGFRa+ expressing cells in the scRNA dataset is needed (see public review). Also included should be some of the key myofibroblast genes (elastin, Acta2, etc.) and their changes in the relevant cell populations. It would be important to show (at least at the transcriptional level) that myofibroblast differentiation is impaired if the author claims that the alveolarization defect is due to functional myofibroblast impairment. Furthermore, Ect2 expression and changes with treatments should be shown for the different cell populations (relevant to Figure 9).

See responses above

(2) The authors stated that they did not find evidence of fibrosis, scarring, and inflammation, but did not provide data to support this statement. Given the importance of at least the inflammation component in BPD, the absence of inflammation needs to be shown, especially in the model using the TGFBR2-cKO mouse, where at least their data show a trend to increased CD45 cell numbers (Figure 2), and upregulated inflammatory upstream regulators (IL10, IFNa, IKBKB, CEBPB upregulated) in the IPA (Figure 3). BAL and/or tissue by flow or IHC have been used to assess different immune cell populations. In terms of evaluation of vascular impairment, the single-cell data set contains endothelial cells, vascular smooth muscle, and pericytes, which allows interrogation following the two different types of injury (hyperoxia cKO TGFbR2) used for the scRNA-seq experiments.

A full characterization of the immune cell or vascular/endothelial cell compartment within our models is beyond the scope of this current study as we were focusing on the shared changes observed within the lung mesenchyme. None of these compartments exist in isolation, so of course there are likely to be correlative and/or causative changes observed in each of the different models which we studied. We did consider further phenotypic analysis of the immune cells by flow cytometry within our different models, but deferred these experiments for future studies. As mentioned earlier we have omitted the reference to “no inflammation”.

(3) The authors should report several litters per experiment and experimental group, mortality in the groups, and if present, visualize using e.g. Caplan-Meyer curves. The switch of the mothers during treatment, the early postnatal injections and treatments, and variability in outcome measures between different litters have to be anticipated. Therefore at least 2 litters, but preferably 3 litters per experiment should be examined, to show reproducibility.

All experiments were conducted with at least 2-3 contemporaneous litters in each treatment group as this was necessary to have enough animals per treatment condition/group to achieve statistical significance. This was essential as all experiments were conducted on the C57BL/6 background where litter sizes are typically 6-8 pups in our colony. We did not encounter any maternal mortality related to hyperoxia exposure while rotating between hyperoxia and normoxia every 48 hrs. Loss of pups in our experiments was mostly due to cannibalism either immediately after birth or from neglect due to failure of cross-fostering.

(4) The reviewer is concerned about using PBS as a control for experiments involving antibody treatment, in this case, 1D 11. The use of an isotype IgG would be the most appropriate and convincing control. In this case, an isotype-matched murine IgG1 control (13C4) has already been generated and is commercially available. While the reviewer does not suggest repeating all experiments, at least one small experiment showing that control IgG does not alter the lung phenotype with hyperoxia when compared with 1D11 would be important.

We appreciate the reviewer’s suggestion and will consider an isotype antibody comparison in future studies. While not directly comparing 1D11 to isotype, we can share data in which we compared PBS to a different antibody. In this experiment, we attempted to use antibody blockade during the first 10 days of life while mice were undergoing hyperoxia treatment to target a specific component of the TGFb pathway. We observed no difference in outcomes either in RA or O2 when comparing PBS to xxx antibody. We cannot share the antibody identity due to intellectual property reasons, however additional studies confirmed that this antibody likely had no impact due to poor in vivo blocking activity.

**Author response image 10. sa4fig10:** 

(5) While inhibited proliferation is one possible explanation for the decrease of PDGFRa expression in the injured mice, there should be consideration of increased and/or premature apoptosis (before the physiologically observed wave P14-P20) as another reason. Also, do the authors propose that only proliferation results in alveolarization impairment, but differentiation plays no significant role here? If that is the case that would mean that there are some fully-differentiated myofibroblasts in the alveolar septa, but not enough to create the multitude of alveolar septal walls. Have the authors evaluated the decrease in secondary alveolar septa formed per alveolar airspace? This measure would give some sense of whether septum initiation was prevented or whether septa were formed, but are structurally abnormal, e.g. due to altered ECM (suspected decrease in Elastin and SMA expression, if myofibroblast differentiation was impaired or cell content suspected decrease in myofibroblasts and increase of other cell types, such as lipofibroblasts).

Apoptosis/cell death are likely to play a role in addition to inhibited proliferation. See violin plots shown above with cell cycle arrest and pro-apoptotic genes upregulated within the mesenchyme. Because we were unable to optimize tissue sections/staining with the samples collected during the early time points of our experiments (ie P4, P7, P10, P14), we are unable to co-stain for markers of apoptosis and answer this question in a direct manner. Future experiments will focus on additional characterization of these early changes with particular attention to altered fibroblast phenotypes within the alveolar septae.

(6) An illustration depicting key cells and the pathways involved in cartoon format would be a useful addition and visualize the important conclusions of this paper for the reader.

We appreciate this suggestion but think the results are sufficiently straightforward that a summary cartoon would not add much.

Figure 4A: the legend appears to be switched. The gray square seems to align with the epithelial ligands, while the blue square aligns with receptors.

Thank you for identifying this mistake – fixed.

Names of transgenic lines used through manuscript:Please use the correct name, as per JAX would be either Gli1tm3(cre/ERT2)Alj/J or Gli1-CreERT2.Please use the correct name, as per JAX would be either Pdgfratm1.1(cre/ERT2)Blh/J or Pdgfrα-CreERT2.PDGFRa-CRE would be JAX# 013148.

The transgenic lines have been noted in the methods, and we have edited the text of the manuscript to reflect the correct names of these lines. For the supplementary figure 4 which compares Gli1-CreERT2 to Pdgfrα-CreERT2, we left our prior nomenclature intact because it better reflects that each of these lines are haploinsufficient at their targeted loci, and that the controls are cre-negative littermates.

We did not use the PDGFRa-CRE line (JAX# 013148).

**Reviewer #3 (Recommendations For The Authors):**
- More transparency about the single-cell analysis is required: (1) how are cell types and clusters defined? (2) what strategy was used for ambient RNA? (3) how do the controls compare with recently published mouse developmental datasets? (4) how does this model compare with the single-cell dataset published by Thibeault et al in 2021 (neonatal hyperoxia x 14 days with multiple time points used)?

See responses above.

- Tissue level validation of these findings is essential by RNA ISH or IF. While validation that the same process is at play in human tissue would be ideal, if this is not available, the conclusions must be tempered in the discussion.

See responses above.

- Is this more mild neonatal injury reversible in mice? As noted above, more characterization of this model (and placing it in the context of other more widely published models would be helpful).

See responses above.